

# Quasi static ensemble variational data assimilation

Anthony Fillion[1,2], Marc Bocquet[1], and Serge Gratton[3]

[1]CEREA, Joint Laboratory École des Ponts ParisTech and EDF R&D, Université Paris-Est, Champs-sur-Marne, France
[2]CERFACS, Toulouse, France
[3]INPT-IRIT, Toulouse, France

*Correspondence to:* Anthony Fillion (anthony.fillion@enpc.fr)

**Abstract.** The analysis in nonlinear variational data assimilation is the solution of a non-quadratic minimization. Thus, the analysis efficiency relies on its ability to locate a global minimum of the cost function. If this minimization uses a Gauss-Newton (GN) method, it is critical for the starting point to be in the attraction basin of a global minimum. Otherwise the method may converge to a *local* extremum, which degrades the analysis. With chaotic models, the number of local extrema often increases with the temporal extent of the data assimilation window, making the former condition harder to satisfy. This is unfortunate because the assimilation performance also increases with this temporal extent. However, a quasi-static (QS) minimization may overcome these local extrema. It consists in gradually injecting the observations in the cost function. This method was introduced by Pires et al. (1996) in a 4D-Var context.

We generalize this approach to four-dimensional nonlinear EnVar methods, which are based on both a nonlinear variational analysis and the propagation of dynamical error statistics via an ensemble. This forces to consider the cost function minimizations in the broader context of cycled data assimilation algorithms. We adapt this QS approach to the iterative ensemble Kalman smoother (IEnKS), an exemplar of nonlinear deterministic 4D EnVar methods. Using low-order models, we quantify the positive impact of the QS approach on the IEnKS, especially for long data assimilation windows. We also examine the computational cost of QS implementations and suggest cheaper algorithms.

## 1 Introduction

Data assimilation (DA) aims at gathering knowledge about the state of a system from acquired observations. In the Bayesian framework, this knowledge can be represented by the posterior probability density function (pdf) of the system state given the observations. A specificity of sequential DA is that observations are not directly available; they become available as time goes by. Thus, the posterior pdf should be regularly updated.

In order to do so, one usually proceeds in two steps: the analysis and the propagation (or forecast). During the analysis step, a background pdf is used as a prior together with the observation likelihood to construct the (often approximate) posterior pdf, following Bayes' theorem. During the propagation step, this posterior pdf is propagated in time with the model to yield the prior pdf of the next assimilation cycle.





In general these posterior and prior pdfs are not easily computable. In the Kalman filter, assumptions are notably made on the linearity of operators, to keep these pdfs Gaussian. This way, they are characterized by their mean and covariance matrix. Linear algebra is then sufficient to enforce both Bayes' theorem and the propagation step into operations on means and covariances.

However, with nonlinear models, the Kalman filter assumptions do not hold. The posterior and prior pdfs are not Gaussian anymore. A possibility in this case is to enforce Gaussianity with approximations. This requires the selection of mean and covariances intended for the Gaussian surrogate pdfs. With the Kullback-Leibler divergence, the best Gaussian approximation of a pdf is achieved by equating the mean and covariances (see, e.g., Bishop, 2006). However, the integrations necessary to evaluate these moments are also prohibitive.

In the 4D-Var algorithm (see, e.g., Lorenc, 2014, and references therein), Laplace's approximation gives us a way to work around the problem by replacing the posterior mean with the presumed unique global minimizer of the cost function. A model propagation is then sufficient to estimate the prior pdf mean. This approach calls for efficient global optimization routines. However, in practice, solving a global optimization problem is challenging when the number of unknowns is large, and local methods like Gauss-Newton are often preferred in practice (see, e.g., Björck, 1996).

Unfortunately, Gauss-Newton methods' ability to locate the global minimum depends on the minimization starting point and on the cost function properties. Furthermore, missing this global minimum is likely to cause a quick divergence (from the truth) of the sequential DA method. Thus, it is critical for the assimilation algorithm to keep the minimization starting point in a global minimum basin of attraction.

This requirement is constraining because, with a chaotic model, the number of local minima may increase exponentially with the data assimilation window (DAW) time extent $L$ (Miller et al., 1994; Pires et al., 1996). Unfortunately, assuming a perfect, chaotic – and hence unstable – model, this is also for the longest time extents that the assimilation performs best. Several strategies have been investigated to go beyond this restriction. Pires et al. (1996) propose the quasi-static (QS) minimization in a 4D-Var context: as the observations are progressively added to the cost function, the starting point (or first guess) of the 4D-Var minimization is also gradually updated. Ye et al. (2015) propose to gradually increase the model error covariances in the weak-constraint 4D-Var cost function in a minimization over an entire trajectory; this way the model nonlinearity is gradually introduced into the cost function (see also Judd et al., 2004). They also propose to parallelize this minimization over multiple starting points to increase the chance to locate the global minimum.

On the one hand, the 4D-Var benefits from the QS approach to approximate the posterior and prior means. On the other hand, with traditional 4D-Var, the prior covariance matrix is taken as static. This is appropriate when, as in Pires et al. (1996), Swanson et al. (1998) or Ye et al. (2015), only one cycle of assimilation is considered. But this limits the dynamical transfer of error statistics from one cycle to the next.

In contrast, 4D ensemble variational (EnVar) schemes allow to both perform a nonlinear variational analysis and a propagation of dynamical errors via the ensemble (see Chapter 7 of Asch et al., 2016). The improvement brought by QS minimizations on these schemes has been suggested and numerically evaluated in Bocquet and Sakov (2013, 2014); Goodliff et al. (2015). This motivates a more complete analytical and numerical investigation.



The iterative ensemble Kalman smoother (IEnKS) (Bocquet and Sakov, 2014; Bocquet, 2016) is the archetype of such 4D nonlinear EnVar scheme, where the ensemble parts of the algorithm are deterministic. Using low-order models (usually toy-models), it was shown to significantly outperform 4D-Var, the EnKF or the ensemble Kalman smoother in terms of accuracy.

The IEnKS improves the DA cycling by keeping track of the pdfs mean and covariance matrix. To do this, Laplace's

approximation is used to replace the posterior mean and covariance matrix with the minimizer of the cost function and an approximation of the inverse Hessian at the minimizer, respectively. These moments are then used to update the ensemble statistics. The updated ensemble is then propagated to estimate the prior mean and covariance matrix. Hence, it is also critical for the IEnKS to locate the global minimum of the cost function.

Here, we are interested in the application of the QS minimization to the IEnKS. It will be shown that the bigger the DAW,

the better the performance as long as the global minimum is found. Because the QS minimization improves this detection, the IEnKS will benefit from it.

The rest of the paper is organized as follows. In section 2, the performance dependency of 4D-Var and IEnKS algorithms on the DAW parameters is investigated. In order to do so, a brief presentation of 4D-Var and the IEnKS algorithms is given. Then we define a measure of performance for assimilation algorithms. This definition is used to give analytic expressions for the

accuracy of both algorithms with a linear, diagonal, autonomous model. This quantifies the impact of cycling on the algorithms. After these preliminaries, the nonlinear, chaotic case is studied. In section 3, we provide and describe the algorithms of the quasi-static IEnKS (IEnKS-QS). Section 4 is dedicated to numerical experiments with two Lorenz low-order models and to the improvement of the IEnKS-QS numerical efficiency. Conclusions are given in section 5.

We emphasize that the algorithmic developments of this study are not meant to improve high-dimensional, imperfect model,

data assimilation techniques. Even if Miller et al. (1994) show some similarities between the perfect and imperfect settings of a model of intermediate complexity, model error would generally forbid the use of very long DAWs as sometimes considered in this study. Instead, the objective of this paper is to better understand the interplay between chaotic dynamics, ensemble variational data assimilation schemes and their cycling, irrespective of whether they could be useful in high-dimensional systems.

## 25  2  The data assimilation window and the assimilation performance

After reviewing 4D-Var (in a constant background matrix version) and the IEnKS algorithms, we will investigate the dependency of assimilation performance on the DAW key parameters. This will illustrate the cycling improvement brought in by the IEnKS compared to 4D-Var. We shall see that, with chaotic models, the longer the DAW is, the better the accuracy of these algorithms.

The evolution and observation equations of the system are assumed of the form:

$$\mathbf{y}_l = \mathcal{H}(\mathbf{x}_l) + \varepsilon_l, \tag{1a}$$

$$\mathbf{x}_{l+1} = \mathcal{M}(\mathbf{x}_l), \tag{1b}$$





where the unknown state $\mathbf{x}_l$ at time $t_l$ is propagated to $t_{l+1}$ with the model resolvent $\mathcal{M} : \mathbb{R}^m \to \mathbb{R}^m$. The model is assumed to be perfect so that there are no errors in Eq. (1b) and autonomous ($\mathcal{M}$ does not depend on time). The observation operator $\mathcal{H} : \mathbb{R}^m \to \mathbb{R}^d$ relates the state $\mathbf{x}_l$ to the observation vector $\mathbf{y}_l$. The observation errors $(\varepsilon_l)_{l \geq 0}$ are assumed Gaussian with mean $\mathbf{0} \in \mathbb{R}^d$ and covariance matrix $\mathbf{R} \in \mathbb{R}^{d \times d}$; they are uncorrelated in time.

## 2.1 4D-Var and IEnKS algorithms

Both 4D-Var and the IEnKS use a variational minimization in their analysis step. The objective of this minimization is to locate the global maximum of the posterior pdf $p(\mathbf{x}_0 | \mathbf{y}_{L:K})$ of the system past state $\mathbf{x}_0$ given the observations $\mathbf{y}_{L:K} = [\mathbf{y}_K, \ldots, \mathbf{y}_L]$ at times $t_{L:K} = [t_K, \ldots, t_L] \in \mathbb{R}^S$. The system state and observations are seen as random vectors with values in $\mathbb{R}^m$ and $\mathbb{R}^d$, respectively. The posterior pdf quantifies how our knowledge on the state $\mathbf{x}_0$ changes with realizations of $\mathbf{y}_{L:K}$. Thus, its maximum is the most probable state after assimilating the observations. The DAW is displayed in Fig. 1. The parameters $K$ and $L$ are the time index of the DAW first and last assimilated observation batch, respectively. The number of observations is $S$ and satisfies $S = L - K + 1$ in our case (i.e., no overlap between the DAWs). To specify this posterior pdf, we have to make

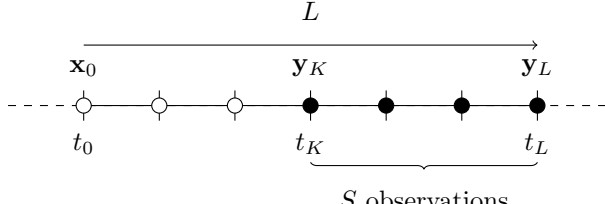

**Figure 1.** Schematic of a DAW. The state variable at $t_0$ is $\mathbf{x}_0$, the observation vector $\mathbf{y}_K$ at time $t_K$ is the first of the DAW to be assimilated and the observation $\mathbf{y}_L$ at present time $t_L$ is the last one, $S = L - K + 1$ is the number of observations to assimilate in the current cycle. These observations, possibly observation vectors, are represented by black dots.

further assumptions on $\mathbf{x}_0$.

The initial state $\mathbf{x}_0$ is assumed to be Gaussian with mean $\mathbf{x}_0^{\mathrm{b}} \in \mathbb{R}^m$ and covariance matrix $\mathbf{B} \in \mathbb{R}^{m \times m}$:

$$p(\mathbf{x}_0) = \mathcal{N}\left(\mathbf{x}_0 | \mathbf{x}_0^{\mathrm{b}}, \mathbf{B}\right). \tag{2}$$

With these assumptions, an analytic expression can be obtained for the posterior pdf $p(\mathbf{x}_0 | \mathbf{y}_{L:K})$ at the first cycle, or for the cost function associated with this pdf. The latter is defined as

$$G(\mathbf{x}_0 | \mathbf{y}_{L:K}) = -\ln p(\mathbf{x}_0 | \mathbf{y}_{L:K}). \tag{3}$$



The notation $G$ is used rather than the traditional $J$ to refer to an exact cost function, i.e., a cost function defined from an exact posterior pdf. Bayes' theorem yields:

$$G\left(\mathbf{x}_0|\mathbf{y}_{L:K}\right) = -\ln p\left(\mathbf{x}_0\right) - \ln p\left(\mathbf{y}_{L:K}|\mathbf{x}_0\right)$$
$$+ \ln p\left(\mathbf{y}_{L:K}\right), \tag{4}$$

and with the Gaussian assumption on the background and observation errors we have

$$G\left(\mathbf{x}_0|\mathbf{y}_{L:K}\right) = \frac{1}{2}\left\|\mathbf{x}_0^b - \mathbf{x}_0\right\|_{\mathbf{B}^{-1}}^2 + c_0$$
$$+ \frac{1}{2}\sum_{l=K}^{L}\left\|\mathbf{y}_l - \mathcal{H}\circ\mathcal{M}^l\left(\mathbf{x}_0\right)\right\|_{\mathbf{R}^{-1}}^2, \tag{5}$$

where $\|\mathbf{x}\|_{\mathbf{A}}^2 = \mathbf{x}^{\mathrm{T}}\mathbf{A}\mathbf{x}$ is, the norm of $\mathbf{x}$ associated with a symmetric positive definite matrix $\mathbf{A}$; $c_0$ is a normalization constant ensuring $\int e^{-G(\mathbf{x}_0|\mathbf{y}_{L:K})}\mathrm{d}\mathbf{x}_0 = 1$; $\mathcal{M}^l$ stands for $l$ compositions of $\mathcal{M}$.

The propagation corresponds to a time shift of $S$ time steps. Thus, at the $k$-th assimilation cycle, the posterior pdf is $p\left(\mathbf{x}_{kS}|\mathbf{y}_{kS+L:K}\right)$. Using Bayes' theorem the $k$-th cycle cost function is:

$$G\left(\mathbf{x}_{kS}|\mathbf{y}_{kS+L:K}\right) = -\ln p\left(\mathbf{x}_{kS}|\mathbf{y}_{(k-1)S+L:K}\right)$$
$$+ \frac{1}{2}\sum_{l=K}^{L}\left\|\mathbf{y}_{kS+l} - \mathcal{H}\circ\mathcal{M}^l\left(\mathbf{x}_{kS}\right)\right\|_{\mathbf{R}^{-1}}^2 + c_{kS}, \tag{6}$$

where $-\ln p\left(\mathbf{x}_{kS}|\mathbf{y}_{(k-1)S+L:K}\right)$ is the background term. If the model and observation operators are nonlinear, an analytical

expression for this prior is not accessible and one needs to approximate it. The 4D-Var and the IEnKS algorithms are solutions based on distinct approximation strategies.

The 4D-Var cost function at the $k$-th cycle, based on the static error covariance matrix $\mathbf{B}$, is defined by

$$J\left(\mathbf{x}_{kS};\mathbf{y}_{kS+L:kS+K},\mathbf{x}_{kS}^{\mathrm{b}}\right) = \frac{1}{2}\left\|\mathbf{x}_{kS}^b - \mathbf{x}_{kS}\right\|_{\mathbf{B}^{-1}}^2$$
$$+ \frac{1}{2}\sum_{l=K}^{L}\left\|\mathbf{y}_{kS+l} - \mathcal{H}\circ\mathcal{M}^l\left(\mathbf{x}_{kS}\right)\right\|_{\mathbf{R}^{-1}}^2. \tag{7}$$

The analysis of 4D-Var consists in minimizing Eq. (7), yielding $\mathbf{x}_{kS}^{\mathrm{a}}$ at $t_{kS}$. Because this cost function depends on realizations of the random observations, $\mathbf{x}_{kS}^{\mathrm{a}}$ is also a random variable. This analysis is then propagated at time $t_{(k+1)S}$ with the resolvent of the model to produce the next cycle background state:

$$\mathbf{x}_{(k+1)S}^{\mathrm{b}} = \mathcal{M}^S\left(\mathbf{x}_{kS}^{\mathrm{a}}\right). \tag{8}$$

In general, $G\left(\mathbf{x}_0|\mathbf{y}_{L:K}\right)$ and $J\left(\mathbf{x}_0;\mathbf{y}_{L:K},\mathbf{x}_0^{\mathrm{b}}\right)$ only coincide at the first cycle of an assimilation because of the assumption

Eq. (2). Subsequently, the background term of the 4D-Var cost function $\frac{1}{2}\left\|\mathbf{x}_{kS}^{\mathrm{b}} - \mathbf{x}_{kS}\right\|_{\mathbf{B}^{-1}}^2$ is a Gaussian approximation of the exact background term $-\ln p\left(\mathbf{x}_{kS}|\mathbf{y}_{(k-1)S+L:K}\right)$. By definition, the background error covariance matrix $\mathbf{B}$ of the traditional 4D-Var cost function is the same for each cycle. This is not the case for the IEnKS.





The IEnKS (Bocquet and Sakov, 2014) is an ensemble method with a variational analysis. Two versions of the algorithm exist: the singular data assimilation (SDA) version where observations are assimilated only once; the multiple data assimilation (MDA) version where they are assimilated several times. We focus on the SDA version in the theoretical development. The MDA version is mentioned in the numerical experiments. At the $k$-th cycle of the IEnKS, the background ensemble at $t_{kS}$ is

5 obtained by a propagation from the previous cycle. The ensemble members are the columns of the matrix $\mathbf{E}_{kS}^{\mathrm{b}}$, which is seen as a random matrix with values in $\mathbb{R}^{m \times n}$. It is used to estimate the prior mean and covariance matrix:

$$\mathbb{E}\left[\mathbf{x}_{kS}|\mathbf{y}_{(k-1)S+L:K}\right] \simeq \bar{\mathbf{x}}_{kS}^{\mathrm{b}}, \tag{9}$$

$$\mathbb{C}\left[\mathbf{x}_{kS}|\mathbf{y}_{(k-1)S+L:K}\right] \simeq \mathbf{X}_{kS}^{\mathrm{b}}\mathbf{X}_{kS}^{\mathrm{bT}}, \tag{10}$$

where $\mathbb{E}$ and $\mathbb{C}$ are the expectation and covariance operators, respectively; $\bar{\mathbf{x}}_{kS}^{\mathrm{b}}$ and $\mathbf{X}_{kS}^{\mathrm{b}}$ are the empirical mean and normalized

anomaly of $\mathbf{E}_{kS}^{\mathrm{b}}$, respectively:

$$\bar{\mathbf{x}}_{kS}^{\mathrm{b}} = \mathbf{E}_{ks}^{\mathrm{b}}\frac{\mathbf{1}_n}{n}, \tag{11}$$

$$\mathbf{X}_{kS}^{\mathrm{b}} = \mathbf{E}_{ks}^{\mathrm{b}}\frac{\mathbf{I}_n - \frac{\mathbf{1}_n\mathbf{1}_n^{\mathrm{T}}}{n}}{\sqrt{n-1}}, \tag{12}$$

with $\mathbf{1}_n = [1,\dots,1]^{\mathrm{T}} \in \mathbb{R}^n$ a vector of ones and $\mathbf{I}_n$ is the identity of $\mathbb{R}^n$. Note that Eqs (9, 10) are not exact because of sampling errors. If the state vector is of the form:

$$\mathbf{x}_{kS} = \bar{\mathbf{x}}_{kS}^{\mathrm{b}} + \mathbf{X}_{kS}^{\mathrm{b}}\mathbf{w}_{kS}, \tag{13}$$

with $\mathbf{w}_{kS} \in \mathbb{R}^n$ the control variable in the ensemble space, the IEnKS cost function is defined in the ensemble space by

$$
\begin{aligned}
J\left(\mathbf{w}_{kS};\mathbf{y}_{kS+L:kS+K},\mathbf{E}_{kS}^{\mathrm{b}}\right) &= \frac{1}{2}\left\|\mathbf{w}_{kS}\right\|^2 \\
&+ \frac{1}{2}\sum_{l=K}^{L}\left\|\mathbf{y}_{kS+l} - \mathcal{H}\circ\mathcal{M}^l\left(\bar{\mathbf{x}}_{kS}^{\mathrm{b}} + \mathbf{X}_{kS}^{\mathrm{b}}\mathbf{w}_{kS}\right)\right\|_{\mathbf{R}^{-1}}^2,
\end{aligned} \tag{14}
$$

where $\|\cdot\| = \|\cdot\|_{\mathbf{I}}$ with $\mathbf{I}$ the identity matrix. The analysis of the IEnKS also consists in minimizing this cost function. It yields

an analyzed ensemble $\mathbf{E}_{kS}^{\mathrm{a}}$ at time $t_{kS}$ verifying:

$$\bar{\mathbf{x}}_{kS}^{\mathrm{a}} = \bar{\mathbf{x}}_{kS}^{\mathrm{b}} + \mathbf{X}_{kS}^{\mathrm{b}}\mathbf{w}_{kS}^{\mathrm{a}}, \tag{15}$$

$$\mathbf{X}_{kS}^{\mathrm{a}} = \mathbf{X}_{kS}^{\mathrm{b}}\left[\nabla^2 J\left(\mathbf{w}_{kS}^{\mathrm{a}};\mathbf{y}_{kS+L:kS+K},\mathbf{E}_{kS}^{\mathrm{b}}\right)\right]^{-1/2}\mathbf{U}, \tag{16}$$

with $\bar{\mathbf{x}}_{kS}^{\mathrm{a}}$ and $\mathbf{X}_{kS}^{\mathrm{a}}$ respectively the empirical mean and normalized anomalies of the analyzed ensemble, $\mathbf{w}_{kS}^{\mathrm{a}}$ the cost function minimizer, $\nabla^2 J\left(\mathbf{w}_{kS}^{\mathrm{a}};\mathbf{y}_{kS+L:kS+K},\mathbf{E}_{kS}^{\mathrm{b}}\right)$ its Hessian at this minimum (usually approximated to avoid computations of the

25 model second derivatives) and $\mathbf{U} \in \mathbb{R}^{n \times n}$ an orthogonal matrix such that $\mathbf{U}\mathbf{1}_n = \mathbf{1}_n$. This analyzed ensemble is also propagated to time $t_{(k+1)S}$ to produce the next cycle background ensemble

$$\mathbf{E}_{(k+1)S}^{\mathrm{b}} = \mathcal{M}^S\left(\mathbf{E}_{kS}^{\mathrm{a}}\right). \tag{17}$$

The cycle is completed by using in the next analysis the cost function Eq. (14) with time indexes incremented by $S$. 4D-Var and the IEnKS are sketched in Fig. 2.





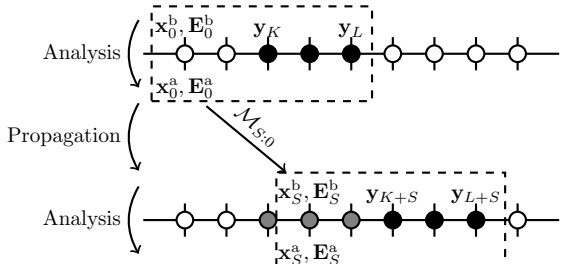

**Figure 2.** Chaining of the 4D-Var and IEnKS two first cycles with $S = 3, L = 4$. The first 4D-Var analysis use the background $\mathbf{x}_0^b$ at $t_0$ and the observations $\mathbf{y}_{L:K}$ to produce the analysis $\mathbf{x}_0^a$ at $t_0$. It is propagated $S$ steps forward in time to produce the new background at time $t_S$ where another analysis can be performed. The IEnKS does the same but with an ensemble. The dashed rectangle symbolizes the current DAW, black dots represent the observations assimilated in the current cycle, gray dots represent already assimilated observations and white dots represent observations not assimilated.

## 2.2 Performance of assimilation

In order to evaluate the efficiency of 4D-Var and the IEnKS with DAW parameters S and L, two measures of accuracy are investigated here: the usual empirical RMSE, and a theoretical counterpart.

At the $k$-th cycle, the algorithm generates at time $t_{kS}$ an analysis $\mathbf{x}_{kS}^a$ from the observations. This analysis is propagated with

the model $l$ steps forward in time to yield the analysis $\mathbf{x}_{kS+l}^a$ meant to approximate the system true state $\mathbf{x}_{kS+l}$. A traditional measure of an assimilation performance is the root mean square error (RMSE). It is defined by

$$\mathrm{RMSE} = \frac{1}{\sqrt{m}} \left\| \mathbf{x}_{kS+l} - \mathbf{x}_{kS+l}^a \right\|. \tag{18}$$

The RMSE takes different names depending on the time it is computed at. If $l = L$, it is called filtering RMSE; if $0 \leq l \leq L-1$, it is the smoothing RMSE with lag $L - l$. In the following, the smoothing RMSE will correspond to the one with (maximum)

lag $L$.

The RMSE rigorously depends on the random variable realizations, and thus it is also a random variable. In our numerical experiments, as is usually done, the RMSE is averaged over the cycles to mitigate this variability:

$$\mathrm{aRMSE}_N = \frac{1}{N} \sum_{k=0}^{N-1} \frac{1}{\sqrt{m}} \left\| \mathbf{x}_{kS+l} - \mathbf{x}_{kS+l}^a \right\|. \tag{19}$$

Let us assume there is a random couple $\left( \mathbf{x}_{\infty S+l}, \mathbf{x}_{\infty S+l}^a \right)$ whose distribution is invariant and ergodic with respect to the shift

transformation:

$$T : \left( \mathbf{x}_{kS+l}, \mathbf{x}_{kS+l}^a \right) \mapsto \left( \mathbf{x}_{(k+1)S+l}, \mathbf{x}_{(k+1)S+l}^a \right). \tag{20}$$

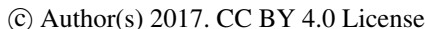



Then by Birkhoff's ergodic theorem (see Walters, 1982) the sequence $(\mathrm{aRMSE}_N)_N$ converges when $N \to \infty$ and its limit aRMSE verifies:

$$\mathrm{aRMSE} = \frac{1}{\sqrt{m}} \mathbb{E}\left[\left\|\mathbf{x}_{\infty S+l} - \mathbf{x}^{\mathrm{a}}_{\infty S+l}\right\|\right], \tag{21}$$

where the expectation $\mathbb{E}$ is taken over $p\left(\mathbf{x}_{\infty S+l}, \mathbf{x}^{\mathrm{a}}_{\infty S+l}\right)$. In this case, the aRMSE measures the long time impact of the cycling

on the assimilation accuracy. This limit is difficult to exploit algebraically. That is why, in the theoretical developments, we will prefer the expected MSE (eMSE), denoted by $P$:

$$P_{kS+l} = \mathbb{E}\left[\left\|\mathbf{x}_{kS+l} - \mathbf{x}^{\mathrm{a}}_{kS+l}\right\|^2\right],$$

where the expectation is taken over $p\left(\mathbf{x}_{kS+l}, \mathbf{x}^{\mathrm{a}}_{kS+l}\right)$. In the following subsection, we will focus on the long term impact of the cycling on $P$. Simplifying assumptions will be made to express $P_{\infty S+l} = \lim_{k \to \infty} P_{kS+l}$ as a function of $S$ and $L$.

## 2.3 Performance in the linear, diagonal, autonomous case

In order to obtain analytical expressions of the eMSE for 4D-Var and the IEnKS, we make drastic simplifying assumptions.

First, the model is assumed to be the resolvent of a linear, diagonal, autonomous ordinary differential equation. Thus, it can be expressed as

$$\mathcal{M}^l\left(\mathbf{x} + \delta\mathbf{x}\right) = \mathcal{M}^l\left(\mathbf{x}\right) + \mathbf{M}^l \delta\mathbf{x}, \tag{22}$$

where $\mathbf{M} = \mathrm{diag}\left(\alpha_i\right)_{i=1..m}$ is diagonal and does not depends on $\mathbf{x}$. We further assume $\mathcal{H} = h\mathbf{I}_m$, $\mathbf{B} = b\mathbf{I}_m$, $\mathbf{R} = r\mathbf{I}_m$, where $\mathbf{I}_m$ is the identity matrix of $\mathbb{R}^m$ and $h, r, b > 0$. With these assumptions, appendix A provides an expression for the 4D-Var asymptotic eMSE in the univariate case. The generalization to the diagonal multivariate case is obtained by summing up the eMSEs of each direction:

$$P^{\text{4D-Var}}_{\infty S+l} = \sum_{i=1}^m \begin{cases} \infty & \text{if } \Delta_i \geq 1 \\ \frac{b^2 \Sigma^L_{K,i}}{\alpha_i^{2(S-l)}} \frac{\Delta_i}{1-\Delta_i} & \text{otherwise} \end{cases}, \tag{23a}$$

$$\Sigma^L_{K,i} = \frac{h^2}{r} \frac{\alpha_i^{2(L+1)} - \alpha_i^{2K}}{\alpha_i^2 - 1}, \tag{23b}$$

$$\Delta_i = \frac{\alpha_i^{2S}}{\left(1 + b\Sigma^L_{K,i}\right)^2}. \tag{23c}$$

The case $\Delta_i \geq 1$ means that too much credit is given to the background variance, which is approximated for all cycles by the constant $b$ in our 4D-Var scheme. Therefore, the information carried by the observations is not sufficient to mitigate the exponential growth of errors in the propagation.



Concerning the IEnKS, the anomalies are assumed to be full rank to avoid any complication due to singular covariance matrices. Moreover, the linearity of the model is employed to express the background statistics:

$$\bar{\mathbf{x}}^{\mathrm{b}}_{(k+1)S} = \mathcal{M}^S(\bar{\mathbf{x}}^{\mathrm{a}}_{kS}),\tag{24a}$$

$$\mathbf{X}^{\mathrm{b}}_{(k+1)S} = \mathbf{M}^S\mathbf{X}^{\mathrm{a}}_{kS},\tag{24b}$$

$$\bar{\mathbf{x}}^{\mathrm{b}}_0 = \mathbf{x}^{\mathrm{b}}_0,\tag{24c}$$

$$\mathbf{X}^{\mathrm{b}}_0 = \mathbf{B}^{1/2}.\tag{24d}$$

This way, sampling errors are avoided as they are not the focus of this study. The background ensemble $\mathbf{E}^{\mathrm{b}}_{kS}$ becomes a notational shortcut for the pair $(\bar{\mathbf{x}}^{\mathrm{b}}_{kS}, \mathbf{X}^{\mathrm{b}}_{kS})$. This simplified IEnKS is a Kalman smoother (Cosme et al., 2012; Bocquet and Carrassi, 2017). With these assumptions, appendix B gives an expression for the IEnKS asymptotic eMSE in the univariate

case. The optimality of the IEnKS eMSE is also proven. The generalization to the diagonal multivariate case is also obtained by summing up the eMSEs of each direction:

$$P^{\mathrm{IEnKS}}_{\infty S+l} = \sum_{i=1}^{m}\begin{cases} 0 & \text{if } |\alpha_i| \le 1 \\ \frac{r}{h^2\alpha_i^{2(L-l)}}\frac{\alpha_i^2-1}{\alpha_i^2} & \text{otherwise} \end{cases}.\tag{25}$$

This expression shows that the eMSE components on the stable directions are null. Indeed one expects the IEnKS to be at least more efficient than a free-run, whose errors in the stable directions tend to zero [1]. This is not the case for 4D-Var since the

15 static background covariance matrix introduces spurious variance in the stable directions as seen in Eq. (23). In Trevisan et al. (2010), 4D-Var error variances in the stable directions are forced to zero to improve the accuracy of the assimilation.

In the following, we study the eMSE dependency on the DAW parameters, $S$ and $L$, for both algorithms. We focus on a bivariate case with $\alpha_1 = 1.2$, $\alpha_2 = 0.8$ in order to have one stable and one unstable direction; $h = b = r = 1$. Using Eqs (23,25), the asymptotic smoothing and filtering eMSEs are displayed as a function of $L, S$ in Fig. 3. The eMSE components on the stable

and unstable directions are also shown. Those graphs are interpreted in the following.

Specifically, the eMSE expression for the IEnKS is of the form

$$P^{\mathrm{IEnKS}}_{\infty S+l} = c_1\alpha_1^{2(l-L)},\tag{26}$$

where $c_1$ does not depend on $S, L, l$. The contribution to $P^{\mathrm{IEnKS}}_{\infty S+l}$ on the stable direction is zero. It depends only on the lag $L-l$; it does not depend on $S$. Moreover, the filtering eMSE ($l = L$) is constant: the propagation compensates for the analysis.

The smoothing eMSE ($l = 0$) decreases exponentially with $L$.

Concerning 4D-Var, we assume $K$ fixed and $S \to \infty$ to give an asymptotic eMSE expression:

$$P^{\mathrm{4D-Var}}_{\infty S+l} = P^{\mathrm{IEnKS}}_{\infty S+l}(1+o(1)) + c_2\alpha_2^{2l}(1+o(1)),\tag{27}$$

where $c_2$ is constant with $S, l$ and $o(1) \to 0$ when $S \to \infty$. The unstable component is close to the IEnKS overall eMSE. The biggest difference with it concerns the eMSE on the stable component. The inexact background variance modeling adds to the

30 eMSE a detrimental term.

[1]The fact that the errors lie in the unstable subspace is more general (Bocquet and Carrassi, 2017)



**Figure 3.** The asymptotic smoothing (on the left column) and filtering (on the right column) eMSEs of 4D-Var and the IEnKS. They are displayed as functions of $L, S$ with their components on the stable, unstable subspaces. The $L$ parameter is on the abscissa axis for the 4D-Var and on the ordinate axis for the IEnKS. The $S$ parameter is on the abscissa axis for the IEnKS and on the ordinate axis for the 4D-Var.

To qualify the long term impact of the cycling on the errors, the filtering eMSE is more instructive than the smoothing eMSE. Indeed, the smoothing eMSE is improved with $L$ as it adds future observations (with respect to the analysis time) in the





DAW. This improvement dominates the detrimental impact of the Gaussian background approximations. In the filtering eMSE, this improvement is balanced by the propagation of the analysis at the end of the DAW. In Fig. 3, the filtering eMSE stable component is damped such that it has little effect. However, on the unstable component, the parameter $S$ improves the filtering eMSE. The bigger $S$, the closer $P_{\infty S+l}^{\mathrm{4D-Var}}$ is to $P_{\infty S+l}^{\mathrm{IEnKS}}$, which is optimal (cf. appendix B). A qualitative explanation is that,

when $S$ is big, the 4D-Var uses less often the inexact background error variance, making the analysis more trustworthy.

Figure 4 displays the asymptotic eMSEs of both algorithms as a function of the lag for $S = L = 5$. These curves are similar

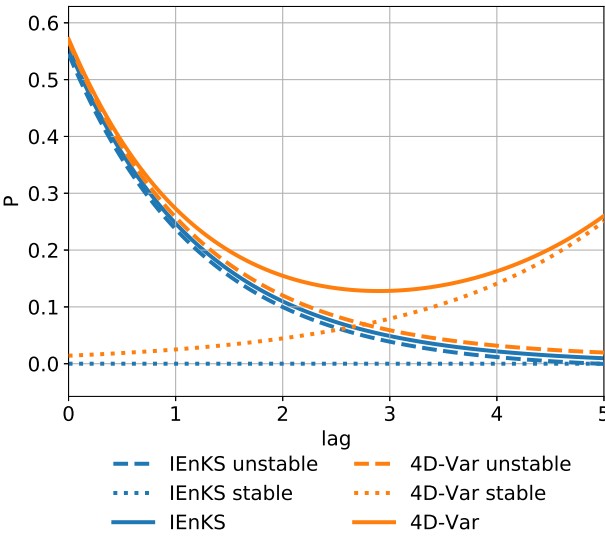

**Figure 4.** The asymptotic eMSEs as a function of the lag (lag $= 0$ is the filtering performance, lag $= L$ is the smoothing performance). The superposing curves have been slightly translated for better readability, $S = L = 5$, $h = b = r = 1$.

to those of Trevisan et al. (2010). Concerning 4D-Var, the eMSE can be written as a function of the lag $L - l$:

$$P_{\infty S+l}^{\mathrm{4D-Var}} \simeq c_1 \alpha_1^{-2\mathrm{lag}} + c_2 \alpha_2^{2(L-\mathrm{lag})}. \tag{28}$$

Thus, the unstable component of the eMSE is an exponentially decreasing function of the lag and the stable component is an

exponentially increasing function of the lag. The sum is therefore decreasing when the unstable component is dominant; it is increasing when the stable component is dominant.

In this section, we have studied the accuracy of both the IEnKS and 4D-Var eMSEs as a function of the DAW parameters under linear, autonomous, diagonal assumptions. We found that the DAW parameter $L$ improves the smoothing eMSE and $S$ improves the filtering eMSE. These properties will be discussed and numerically investigated in a nonlinear context in the next

section.





## 2.4 Performance in the nonlinear, chaotic case

The results of section 2.3 foster the use of the largest possible $L$ to improve the 4D-Var and IEnKS smoothing eMSEs. For filtering, the error propagation at the end of the DAW balances the gain in eMSE due to the assimilation of future observations. Thus, the filtering eMSE is not improved by the assimilation of observations distant in time; it is rather affected by

5 the background pdf approximation. To assimilate the same number of observations, algorithms using high values of $S$ need fewer cycles. Therefore, they rely less often on the background approximation. That is why the 4D-Var filtering performance is improved with $S$. For the IEnKS as in section 2.3, the prior pdf approximation is exact so that there is no dependence on $S$. This is no longer true in the nonlinear case. Hence, with a similar reasoning, one actually expects an improvement on the IEnKS filtering performance with $S$. As a matter of fact, it has been shown that with a nonlinear chaotic model, the filtering

accuracy increases with $L$ in most cases (see Bocquet and Sakov, 2014, and section 4 of the present paper).

However, with a chaotic model, Pires et al. (1996) showed that the 4D-Var cost function number of local extrema increases with $L$, making minimization problematic. We show in this section that the IEnKS cost function suffers from the same problem.

This behavior will be illustrated with the Lorenz 95 (L95) model (Lorenz and Emanuel, 1998). It represents a mid-latitude zonal circle of the atmosphere and is described by a set of $m$ nonlinear differential equations:

$$\frac{\mathrm{d}x_j}{\mathrm{d}t} = (x_{j+1} - x_{j-2})\,x_{j-1} - x_j + F, \tag{29}$$

where $x_j$ is the $j$-th modulo $m$ component of $\mathbf{x}$, $m = 40$ and $F = 8$. This equation is integrated using a fourth-order Runge-Kutta scheme with a time-step of $\delta t = 0.05$. The dynamics of L95 are chaotic; the L95 largest Lyapunov exponent is $\lambda \simeq 1.7$.

Figure 5 shows a typical IEnKS cost function profile in one direction of the analyzed ensemble space for multiple values of the DAW parameters. The system is observed at every time step and $\mathcal{H} = \mathbf{B} = \mathbf{R} = \mathbf{I}_m$. The curves have more and more local

extrema when $L$ increases. The curves with the highest amplitudes of the ripples are found for small values of $S$. Indeed, an averaging effect may settle in as the number of observations increases.

This hilly shape causes minimization problems. A possible minimization procedure for the IEnKS is the Gauss-Newton (GN) algorithm (e.g., Björck, 1996). GN is not a global procedure meaning that, depending on the starting point and the cost function properties, the algorithm can converge towards any local extremum, take many iterations or even diverge. However, if

the cost function is quadratic, the global minimum is reached in one iteration. The cost function non-quadraticity is induced by the model nonlinearity. In the following, we will give a heuristic argument yielding a bound on the $S$ parameter beyond which the GN method probably misses the global minimum in the configuration $\mathcal{H} = \mathbf{B} = \mathbf{R} = \mathbf{I}_m$.

First, the GN convergence properties are drastically simplified. We assume the method converges to the global minimum if and only if the minimization starting point is in a neighborhood of the global minimizer where the IEnKS cost function is

30 almost quadratic.

Unfortunately, this minimizer is unknown because the cost function depends on realizations of many random variables. In order to eliminate this variability, Pires et al. (1996) introduced a so-called error-free cost function. We will rather use an





**Figure 5.** Cost functions of the IEnKS projected in one direction of the analyzed ensemble (hence centered and normalized) with various DAW parameters $S, L$. A quasi-static minimization can be visualized from these panels. It begins with the bottom-left cost function; the orange dot is the starting point and the green one is at the minimum. From the bottom-left cost function up to the top-right cost function, batches of 9 then 10 observation vectors are progressively added to the DAW, and the minimizer (green dot) is updated accordingly.

averaged cost function $J_{\infty S}$ defined by

$$J_{\infty S}(\mathbf{w}) = \lim_{N \to \infty} \frac{1}{N} \sum_{k=0}^{N-1} J\left(\mathbf{w}; \mathbf{y}_{kS+L:kS+K}, \mathbf{E}_{kS}^{\mathrm{b}}\right). \tag{30}$$





Relying on an ergodicity assumption, appendix C proves that this averaged cost function verifies

$$J_{\infty S}(\mathbf{w}) = \frac{1}{2}\|\mathbf{w}\|^2 + \frac{dS}{2} + \frac{1}{2}\sum_{l=K}^{L}\mathbb{E}\left[\left\|\delta\mathbf{x}_{\infty S+l}^{\mathrm{b}}\right\|^2\right], \tag{31a}$$

$$\delta\mathbf{x}_{\infty S+l}^{\mathrm{b}} = \mathcal{M}^l\left(\bar{\mathbf{x}}_{\infty S}^{\mathrm{b}} + \mathbf{X}_{\infty S}^{\mathrm{b}}\mathbf{w}\right) - \mathcal{M}^l\left(\mathbf{x}_{\infty S}\right), \tag{31b}$$

where the ergodic random variables $\bar{\mathbf{x}}_{\infty S}$, $\mathbf{X}_{\infty S}^{\mathrm{b}}$ and $\mathbf{x}_{\infty S}$ have been defined in section 2.2 and Appendix C. As seen in Eq. (31),
a sufficient condition for the starting point $\mathbf{w} = \mathbf{0}$ to be in a neighborhood of the global minimizer where the cost function is assumed almost quadratic is to require that $\bar{\mathbf{x}}_{\infty S}^{\mathrm{b}}$ be in a neighborhood of $\mathbf{x}_{\infty S}$ where all the $\left(\mathcal{M}^l\right)_{K\leq l\leq L}$ are almost linear.

In the univariate case, if the model behavior is almost linear and unstable, we can use Eq. (25) to estimate the terms in the sum in Eq. (31) at the starting point $\mathbf{w} = \mathbf{0}$:

$$\mathbb{E}\left[\left\|\delta\mathbf{x}_{\infty S+l}^{\mathrm{b}}\right\|^2\right] \simeq \alpha^{2(S+l-L)}\frac{\alpha^2 - 1}{\alpha^2}, \tag{32}$$

where $\alpha$ is the model linear part and the extra $S$ accounts for the propagation. But in a necessarily bounded physical system, the right-hand side of Eq. (32) cannot grow indefinitely with $l + S$. Such model saturation imposes

$$\mathbb{E}\left[\left\|\delta\mathbf{x}_{\infty S+l}^{\mathrm{b}}\right\|^2\right] \leq B, \tag{33}$$

with $B$ a bound. Hence, Eqs (32,33) yield the following inequality on $S$:

$$S \leq S_{\max}, \tag{34a}$$

$$S_{\max} = \frac{\ln(B) - \ln\left(1 - \alpha^{-2}\right)}{2\ln(\alpha)}. \tag{34b}$$

We choose $l = L$ because it corresponds to the most constraining case. When Eq. (34a) is violated, $\bar{\mathbf{x}}_{\infty S}^{\mathrm{b}}$ departs from $\mathbf{x}_{\infty S}$ such that the nonlinearities of $\mathcal{M}^L$ are significant.

To apply this inequality to the L95 model we choose:

$$\alpha = \lim_{N\to\infty}\frac{1}{N}\sum_{k=0}^{N-1}\sigma\left(\frac{\mathrm{d}\mathcal{M}}{\mathrm{d}\mathbf{x}}(\mathbf{x}_k)\right), \tag{35}$$

$\sigma$ being the mean of the singular values greater than 1. This corresponds to an average of the error amplification by $\frac{\mathrm{d}\mathcal{M}}{\mathrm{d}\mathbf{x}}(\mathbf{x}_{\infty S})$ in the local unstable subspace. We also choose for the bound $B$ the average squared norm between two long trajectories:

$$B = \lim_{N\to\infty}\frac{1}{mN}\sum_{k=0}^{N-1}\left\|\mathbf{x}_{kS} - \mathcal{M}^{kS}\left(\bar{\mathbf{x}}_0^{\mathrm{b}}\right)\right\|^2. \tag{36}$$

This quantity is greater than $\mathbb{E}\left[\left\|\delta\mathbf{x}_{\infty S+l}^{\mathrm{b}}\right\|^2\right]$ because the IEnKS asymptotic performance is at least better than a free run. From the values of $\alpha$ and $B$ we find $S_{\max} = 14$.

Figure 6 shows the filtering and smoothing aRMSEs of an IEnKS $L = S$ with L95 as a function of $S$, the performance strongly deteriorates for $S > 16$, which is remarkably consistent with our estimation. Figure 6 also shows another difference





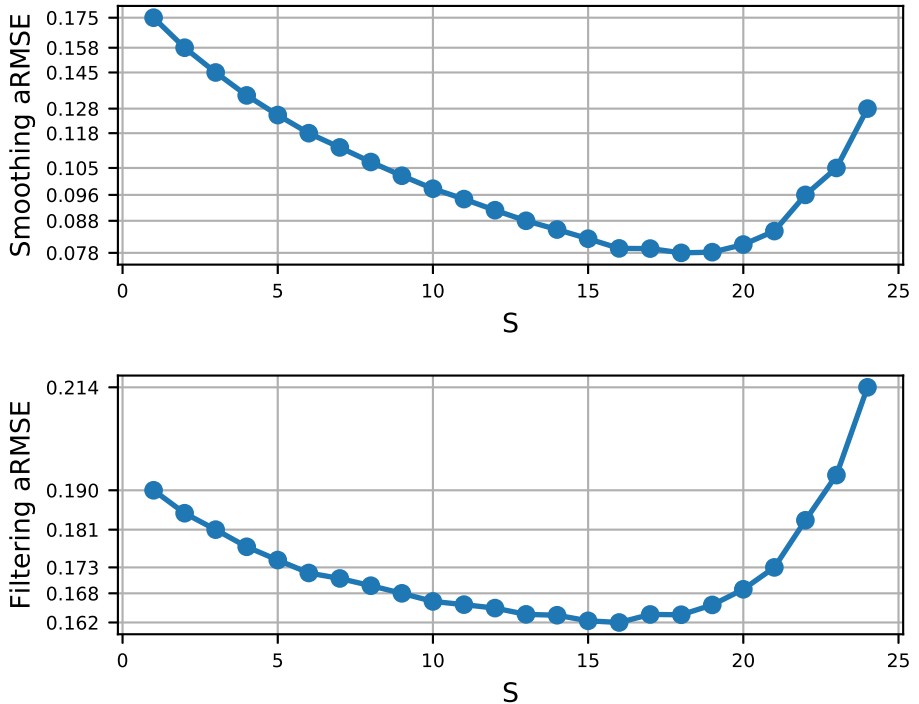

**Figure 6.** Smoothing aRMSE (top) and filtering aRMSE (bottom) of a Gauss-Newton IEnKS $L = S$ as a function of $S$ with L95 model over $5 \times 10^5$ cycles (logarithmic scale). We used the finite-size version (Bocquet, 2011) to account for sampling errors and avoid the need for inflation, $\mathcal{H} = \mathbf{R} = \mathbf{B} = \mathbf{I}_m, n = 20$.

with the linear case: the IEnKS filtering aRMSE depends on $S$. The former discussion on the local extrema explains this dependency for large values of $S$.

However, for small values of $S$, the decreasing aRMSE has not been explained. This is a consequence of the Gaussian background approximation. At each cycle, the IEnKS uses the background ensemble $\mathbf{E}_{kS}^{b}$ to estimate the first two moments of the background pdf and makes the approximation:

$$p\left(\mathbf{x}_{kS}|\mathbf{y}_{(k-1)S+L:K}\right) \simeq \mathcal{N}\left(\mathbf{x}_{kS}|\bar{\mathbf{x}}_{kS}^{b}, \mathbf{X}_{kS}^{b}\left(\mathbf{X}_{kS}^{b}\right)^{\mathrm{T}}\right). \tag{37}$$

Because the model is nonlinear, this pdf is unlikely to be Gaussian. Therefore, this approximation results in a loss of information and the more it is used, the farther from $G$ the IEnKS cost function is. This is exactly what happens when $S$ is small: to assimilate the same number of observations, the IEnKS uses more cycles so that it relies more on the Gaussian background approximation.





## 3 Quasi-static algorithms

We have seen in the previous section that the effective DAW length should be limited by the cost function non-quadraticity. In this section we review and propose algorithms able to overcome these minimization issues and reach longer DAWs.

Quasi static algorithms have been introduced by Pires et al. (1996) in a 4D-Var context. The idea behind quasi-staticism is

to control the way observations are added to the cost function in order to keep the minimization starting point in the basin of attraction containing the global minimizer. The method consists in repeating the minimization with increasing parameter $S$: the first minimization is performed using the cost function with $S = 1$ then $S$ is increased and another minimization can be performed with the former minimizer as the new starting point. The process is then repeated until $S$ reaches its final value.

This procedure is directly applicable to the IEnKS cost function minimization. The left panel in Fig. 7 is a schematic of a QS

minimization and Algorithm 1 gives the pseudo-code of an IEnKS with a QS minimization. The new parameters $(L_q)_{q<N_Q}$ control the number of observations added at each minimization. The three first lines initialize the minimization starting point, the ensemble mean and anomaly matrix. The **for** loop in lines 4-23 repeats the QS minimization. The **while** loop in lines 6-22 is the Gauss-Newton minimization. Lines 7 and 8 center the ensemble on the current minimizer. Lines 9 and 10 initialize the cost function gradient and the approximate Hessian. The **for** loop in lines 12-18 construct observation terms of the cost function

gradient and approximate Hessian. Lines 13 and 14 use a finite difference formula to compute the tangent linear and adjoint of $\mathbf{w} \mapsto \mathcal{H} \circ \mathcal{M}^l \left( \bar{\mathbf{x}}_0^{\mathrm{b}} + \mathbf{X}_0^{\mathrm{b}} \mathbf{w} \right)$. Lines 15 and 16 use this adjoint to update the gradient and approximate Hessian. Lines 19 and 20 solve the linear system of the Guauss-Newton algorithm to update the current minimizer. When GN convergence is reached, this minimizer will be used as a starting point for the next QS minimization. Line 24 updates the ensemble. Line 25 propagates the updated ensemble to the next assimilation cycle.

In Fig. 5, the QS scheme corresponds to minimizing a cost function in the bottom row then using its minimum as a starting point for the minimization of the top right cost function and so on.

Note that Bocquet and Sakov (2014) already qualified the SDA IEnKS with $S = 1$ as QS. The reason is that, at each propagation, only one vector of observations enters and only one leaves the DAW, slightly deforming the cost function. It is also the easiest way to ensure inequality (34a). However, this method has been shown to be suboptimal because of the Gaussian

background approximations.

An alternative to keep $S = 1$ with many observations in the DAW is to relax the condition $K = L - S + 1$. This way, observations are assimilated several times. This is done in the MDA IEnKS (Bocquet and Sakov, 2014). But to keep the statistics consistent at least in the linear/Gaussian case, the observations error covariances are adequately altered. Because $S = 1$, the cost function is slightly modified between each assimilation. That is why the MDA IEnKS is qualified as QS.

However, the multiple assimilation of the observations introduces spurious correlations in the nonlinear/non-Gaussian case, which entail sub-optimality. A scheme similar to the IEnKS-QS has also been successfully used in Carrassi et al. (2017) to compute model evidence. Indeed, the efficient computation of model evidence as an integral over the state space depended on the proper identification of a global maximum of the integrand. However, its implementation was based on the update of





---

**Algorithm 1** One cycle of the IEnKS-QS

---

**Require:** $\mathbf{E}_0^{\mathrm{b}}$ the background ensemble at $t_0$; $\lambda$ the inflation; $(L_q)_{q<N_Q}$ a list of DAW time indexes; $\varepsilon$ the finite differences step; $\delta, j_{\max}$
  GN end of loop parameters; $\mathbf{1} = (1, \ldots, 1)^{\mathrm{T}} \in \mathbb{R}^n$ and $\mathbf{I}$ is the identity of $\mathbb{R}^n$.

1:  $\mathbf{w}_0^{\mathrm{a}} := \mathbf{0}$

2:  $\bar{\mathbf{x}}_0^{\mathrm{b}} = \mathbf{E}_0^{\mathrm{b}} \mathbf{1}/n$

3:  $\mathbf{X}_0^{\mathrm{b}} = \lambda \left( \mathbf{E}_0^{\mathrm{b}} - \bar{\mathbf{x}}_0^{\mathrm{b}} \mathbf{1}^{\mathrm{T}} \right)$

4:  **for** $q = 0 \ldots N_Q - 1$ **do**

5:     $j := 0$

6:     **repeat**

7:        $\bar{\mathbf{x}}_0^{\mathrm{a}} = \bar{\mathbf{x}}_0^{\mathrm{b}} + \mathbf{X}_0^{\mathrm{b}} \mathbf{w}_0^{\mathrm{a}}$

8:        $\mathbf{E}_0 := \bar{\mathbf{x}}_0^{\mathrm{a}} \mathbf{1}^{\mathrm{T}} + \epsilon \mathbf{X}_0^{\mathrm{b}}$

9:        $\nabla J := (n-1) \mathbf{w}_0^{\mathrm{a}}$

10:       $\tilde{\nabla}^2 J := (n-1) \mathbf{I}$

11:       $\mathbf{E}_K := \mathcal{M}^K \left( \mathbf{E}_0 \right)$

12:       **for** $l = K, \ldots, L_q$ **do**

13:          $\bar{\mathbf{y}}_l := \mathcal{H} \left( \mathbf{E}_l \right) \mathbf{1}/n$

14:          $\mathbf{Y}_l := \left( \mathcal{H} \left( \mathbf{E}_l \right) - \bar{\mathbf{y}}_l \mathbf{1}^{\mathrm{T}} \right)/\epsilon$

15:          $\nabla J := \nabla J - \mathbf{Y}_l^{\mathrm{T}} \mathbf{R}^{-1} \left( \mathbf{y}_l - \bar{\mathbf{y}}_l \right)$

16:          $\tilde{\nabla}^2 J := \tilde{\nabla}^2 J - \mathbf{Y}_l^{\mathrm{T}} \mathbf{R}^{-1} \mathbf{Y}_l$

17:          $\mathbf{E}_{l+1} := \mathcal{M} \left( \mathbf{E}_l \right)$

18:       **end for**

19:       solve $\tilde{\nabla}^2 J \delta \mathbf{w} := \nabla J$

20:       $\mathbf{w}_0^{\mathrm{a}} := \mathbf{w}_0^{\mathrm{a}} - \delta \mathbf{w}$

21:       $j := j + 1$

22:     **until** $\|\delta \mathbf{w}\| \leq \delta$ or $j \geq j_{\max}$

23:  **end for**

24:  $\mathbf{E}_0^{\mathrm{a}} := \mathbf{x}_0^{\mathrm{a}} \mathbf{1}^{\mathrm{T}} + \sqrt{n-1} \mathbf{X}_0^{\mathrm{b}} \tilde{\nabla}^2 J^{-1/2}$

25:  $\mathbf{E}_S^{\mathrm{b}} := \mathcal{M}^S \left( \mathbf{E}_0^{\mathrm{a}} \right)$

---

the ensemble whenever an observation batch is added to the cost function, which is not as numerically efficient as the scheme presented here.

The success of the QS minimization lies in the fact that, when an observation is successfully assimilated, the eMSE is reduced. Thus, the analysis probability mass concentrates around the true state. The analysis is then more likely to be in a neighborhood of the true state where the model is linear. The cost function non-quadraticity can then be increased by adding a new term in it. This is confirmed by the following argument. Let $P^{(q)}$ be the IEnKS asymptotic eMSE at the $q$-th step of a QS




minimization. With the notations and assumptions of section 2.3, i.e. in a linear context, we have the recurrence relation:

$$\left(P^{(q+1)}\right)^{-1} = \left(P^{(q)}\right)^{-1} + \Sigma_{L_q+1}^{L_{q+1}}, \tag{38a}$$

$$P^{(-1)} = \alpha^{2S} \frac{\alpha^2 - 1}{\alpha^{2\left(L_{N_Q-1}+1\right)}}. \tag{38b}$$

Thus, $P^{(q+1)} < P^{(q)}$ and we can increase the cost function non-quadraticity by adding new terms in it as long as the propaga-
tion of errors does not exceed the bound

$$\alpha^{2L_{q+1}} P^{(q)} \leq B. \tag{39}$$

This implies

$$L_{q+1} \leq L_q + S_{\max}, \tag{40a}$$

$$L_0 \leq L_{N_Q-1} + S_{\max} - S, \tag{40b}$$

which yields $S \leq N_Q S_{\max}$. Therefore, the QS minimization allows for a $N_Q$ times longer DAW.

Unfortunately, these QS minimizations are very expensive. Indeed, they add a third outer loop repeating $N_Q$ GN minimiza-
tions. The GN iterations used to compute the intermediate starting points give unnecessary precision; all that is required for
these starting points is to be in a neighborhood of $\mathbf{x}_{\infty S}$ where the model is almost linear. Thus, one can restrain the number
of intermediate GN loops and save the full convergence to the last minimization. This is done in the *quasi-convergent* IEnKS
(IEnKS-QC) with the parameters $(j_q)_{q<N_Q}$. They correspond to the numbers of GN loops in the intermediate QS minimiza-
tions. They are typically equal to 1 except for the last one. Algorithm 2 gives the pseudo code of the IEnKS-QC and the right
panel in Fig. 7 is a schematic for it.

## 4   Numerical experiments with low-order models

In the following, we perform numerical experiments with the Lorenz 1963 (L63) and Lorenz 1995 (L95) models. L95 has
already been presented in section 2.4. L63 (Lorenz, 1963) is a simplified model for atmospheric convection. It is defined by
the ordinary differential equations:

$$\frac{\mathrm{d}x}{\mathrm{d}t} = \sigma\left(y - x\right), \tag{41a}$$

$$\frac{\mathrm{d}y}{\mathrm{d}t} = \rho x - y - xz, \tag{41b}$$

$$\frac{\mathrm{d}z}{\mathrm{d}t} = xy - \beta z. \tag{41c}$$

These equations are integrated using a fourth-order Runge-Kutta scheme with a time-step of $\delta t = 0.01$ and $(\sigma, \rho, \beta) = (10, 28, 8/3)$.
The dynamics of the L63 model are chaotic, with a largest Lyapunov exponent given by $\lambda \simeq 0.91$.

Both models are assumed perfect. The truth run is generated from a random state space point. The initial ensemble is
generated from the truth with $\mathbf{B} = \mathbf{I}_m$ where $m = 40, 3$ and $n = 20, 3$ for L95 and L63, respectively. Observation vectors



---

**Algorithm 2** One cycle of the IEnKS-QC

---

**Require:** $\mathbf{E}_0^b$ the background ensemble at $t_0$; $\lambda$ the inflation; $(L_i)_{q<N_Q}$ a list of DAW time indexes; $(j_q)_{q<N_Q}$ the number of intermediate

GN loops; $\varepsilon$ the finite differences step; $\delta$ GN end of loop parameter; $\mathbf{1} = (1,\ldots,1)^T \in \mathbb{R}^n$ and $\mathbf{I}$ is the identity of $\mathbb{R}^n$;

1:    $\mathbf{w}_0^a := \mathbf{0}$

2:    $\bar{\mathbf{x}}_0^b = \mathbf{E}_0^b \mathbf{1}/n$

3:    $\mathbf{X}_0^b = \lambda \left( \mathbf{E}_0^b - \bar{\mathbf{x}}_0^b \mathbf{1}^T \right)$

4:    **for** $q = 0 \ldots N_Q - 1$ **do**

5:      $j := 0$

6:      **repeat**

7:        $\bar{\mathbf{x}}_0^a := \bar{\mathbf{x}}_0^b + \mathbf{X}_0^b \mathbf{w}_0^a$

8:        $\mathbf{E}_0 := \bar{\mathbf{x}}_0^a \mathbf{1}^T + \epsilon \mathbf{X}_0^b$

9:        $\nabla J := (n-1)\mathbf{w}_0^a$

10:       $\tilde{\nabla}^2 J := (n-1)\mathbf{I}$

11:       $\mathbf{E}_K := \mathcal{M}^K (\mathbf{E}_0)$

12:       **for** $l = K, \ldots, L_q$ **do**

13:          $\bar{\mathbf{y}}_l := \mathcal{H}(\mathbf{E}_l)\mathbf{1}/n$

14:          $\mathbf{Y}_l := \left( \mathcal{H}(\mathbf{E}_l) - \bar{\mathbf{y}}_l \mathbf{1}^T \right)/\epsilon$

15:          $\nabla J := \nabla J - \mathbf{Y}_l^T \mathbf{R}^{-1} (\mathbf{y}_l - \bar{\mathbf{y}}_l)$

16:          $\tilde{\nabla}^2 J := \tilde{\nabla}^2 J - \mathbf{Y}_l^T \mathbf{R}^{-1} \mathbf{Y}_l$

17:          $\mathbf{E}_{l+1} := \mathcal{M}(\mathbf{E}_l)$

18:       **end for**

19:       solve $\tilde{\nabla}^2 J \delta\mathbf{w} := \nabla J$

20:       $\mathbf{w}_0^a := \mathbf{w}_0^a - \delta\mathbf{w}$

21:       $j := j + 1$

22:      **until** $\|\delta\mathbf{w}\| \leq \delta$ or $j \geq j_q$

23:   **end for**

24: $\mathbf{E}_0^a := \mathbf{x}_0^a \mathbf{1}^T + \sqrt{n-1}\mathbf{X}_0^b \tilde{\nabla}^2 J^{-1/2}$

25: $\mathbf{E}_S^b := \mathcal{M}^S (\mathbf{E}_0^a)$

---

are generated from the truth with $\mathcal{H} = \mathbf{R} = \mathbf{I}_m$ every $\Delta t = 0.05$ for L95 and every $\Delta t = 0.02$ for L63. A burn-in period of $5 \times 10^3 \times \Delta t$ is enforced in both cases.

     The algorithms parameters are $\varepsilon = 10^{-4}$, $\delta = 10^{-3}$, $j_{max} = 20$, $N_Q = 1$, $L_0 = L$ for the IEnKS. For the IEnKS-QS, the QS parameters are $(L_0, L_1, \ldots, L_{N_Q-1}) = K + (0, 1, \ldots, N_Q - 1) \times \frac{S-1}{N_Q-1}$. For the IEnKS-QC, the QS parameters are the same and $(j_0, \ldots, j_{N_Q-2}, j_{N_Q-1}) = (1, \ldots, 1, 20)$. Sampling errors are systematically accounted for using the IEnKS finite-size version (Bocquet, 2011; Bocquet and Sakov, 2012; Bocquet et al., 2015) which avoids the need for inflation and its costly tuning. Finally the aRMSE is averaged over a number of cycles which is determined by the number of observations assimilated. We use $5 \times 10^5$ observation vectors for L95 and $5 \times 10^6$ observation vectors for L63. Unlike Goodliff et al. (2015), our numerical



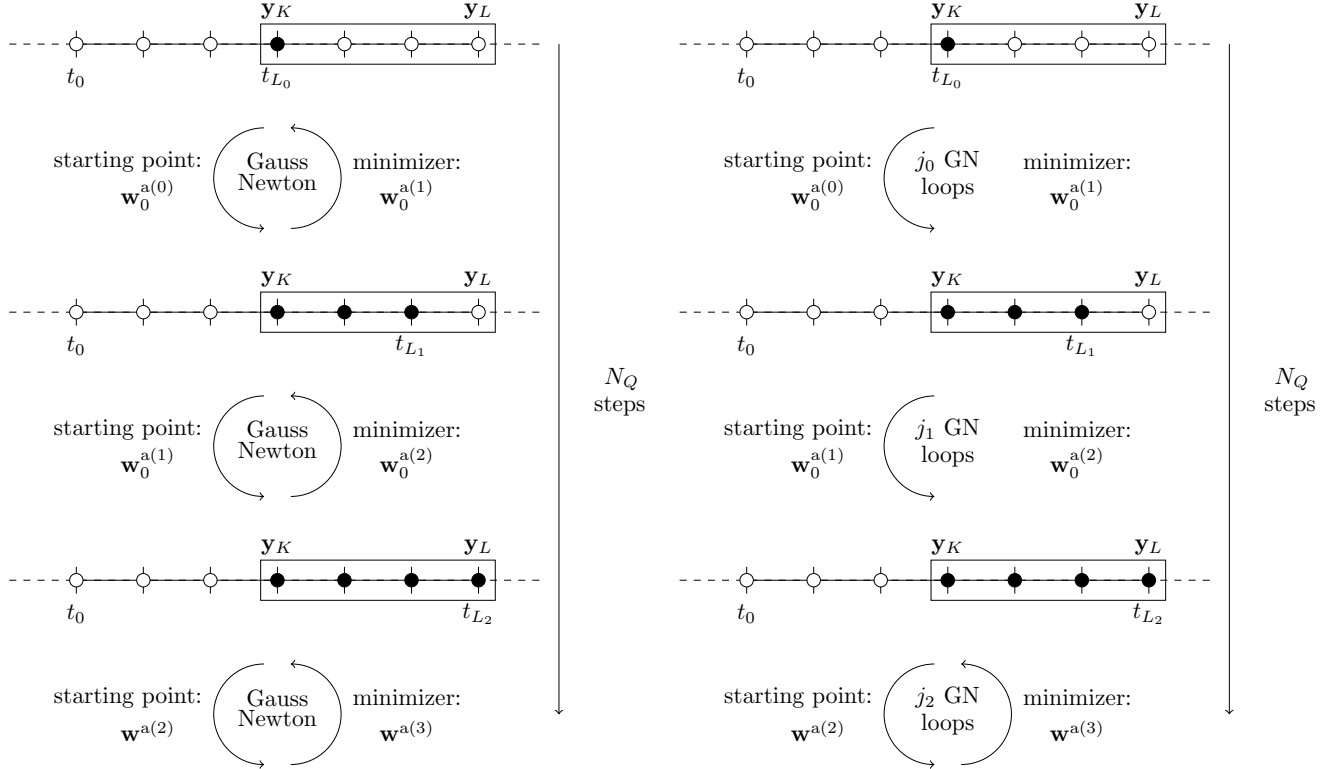

**Figure 7.** Schematics of an IEnKS-QS minimization (on the left) and a IEnKS-QC minimization (on the right). The rectangle contains the observations to be assimilated, the black dots represent the observations used in the current minimization. The "Gauss-Newton" surrounded by arrows represents the iterations of the Gauss-Newton procedure. The number of quasi-static steps is $N_Q = 3$. The flow of observations is controlled by the parameters $(L_0, L_1, L_2) = (3, 5, 6)$. For the QC IEnKS, the number of GN iterations is controlled by the $(j_0, j_1, j_2)$ parameters.

experiments neither address increasing nonlinearity, nor do they adress the use of climatological background error covariance matrices. Instead, we focus exclusively on the IEnKS performance dependence on the DAW key parameters $L$, $S$ and the number $N_Q$ of QS minimizations.

Figures 8, 9 show the aRMSE of the IEnKS and IEnKS-QS for both L63 and L95 as a function of $S$ and $L$. The QS variant allows to reach much longer DAWs, and improves the performance. However, the limit of this method is visible with the L95 model when, for $L = 50$, best aRMSEs are reached for $S < 50$.

Figure 10 compares the smoothing aRMSE, the filtering aRMSE and the number of ensemble propagations of the IEnKS-QS ($N_Q = S, S = L$) and the IEnKS-MDA ($S = 1$), for both L63 and L95 as a function of $L$. For $L < 40$, the IEnKS-QS has smaller aRMSEs than the IEnKS-MDA. The IEnKS-QS is SDA so it does not suffer from suboptimality related to multiple assimilations. Moreover, the IEnKS-QS always requires less propagations of the ensemble which improves the computational





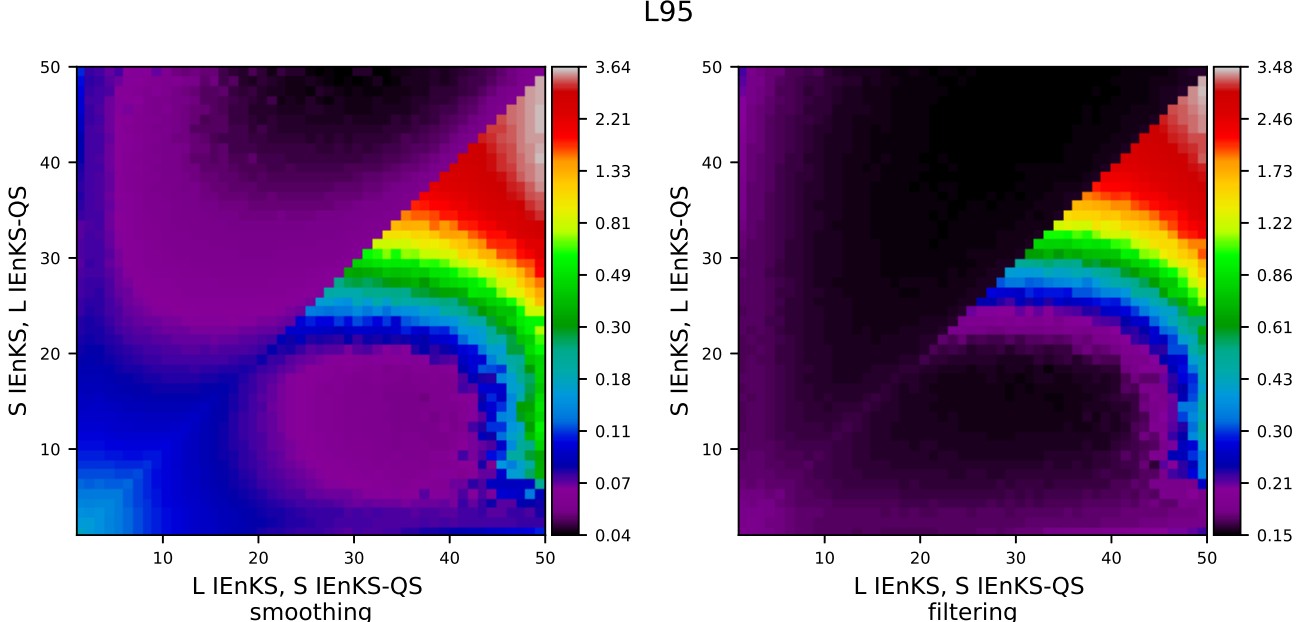

**Figure 8.** IEnKS (lower triangles) and IEnKS-QS (upper triangles, $N_Q = S$) smoothing and filtering aRMSEs as a function of $L$ and $S$ with the L95 model. The $L$ parameter is on the abscissa axis for the IEnKS and on the ordinate axis for the IEnKS-QS. The $S$ parameter is on the abscissa axis for the IEnKS-QS and on the ordinate axis for the IEnKS.

cost. However, for $L > 40$ with the L95 model, the quasi-static approach cannot sustain the non-linearity anymore and the IEnKS-QS aRMSE degrades. Hence, the IEnKS-MDA $S = 1$ has still the best performance.

Figure 11 compares the smoothing aRMSE, the filtering aRMSE and the number of ensemble propagations of the IEnKS-QS and IEnKS-QC as a function of the $N_Q$ parameter. The IEnKS-QS aRMSE decreases quickly after a point for both algorithms and for both models. For the IEnKS-QS with the L95 model, this point can be estimated using results of section 3 by $S/S_{\max} \simeq$ 3.6, in remarkable agreement with the experiments. This point comes later for the IEnKS-QC but it demands less ensemble propagations making this algorithm numerically more efficient than the IEnKS-QS.

## 5 Conclusions

In this paper, we have extended the study of Pires et al. (1996) on quasi-static variational data assimilation, focused on 4D-Var technique, to cycled data assimilation schemes and specifically four-dimensional nonlinear ensemble variationnal techniques, an exemplar of which being the iterative ensemble Kalman smoother (IEnKS).

The long term impact of cycling has been first investigated theoretically in a linear context for 4D-Var and the IEnKS, then numerically for the IEnKS in a nonlinear context. The way information is propagated between data assimilation cycles indeed





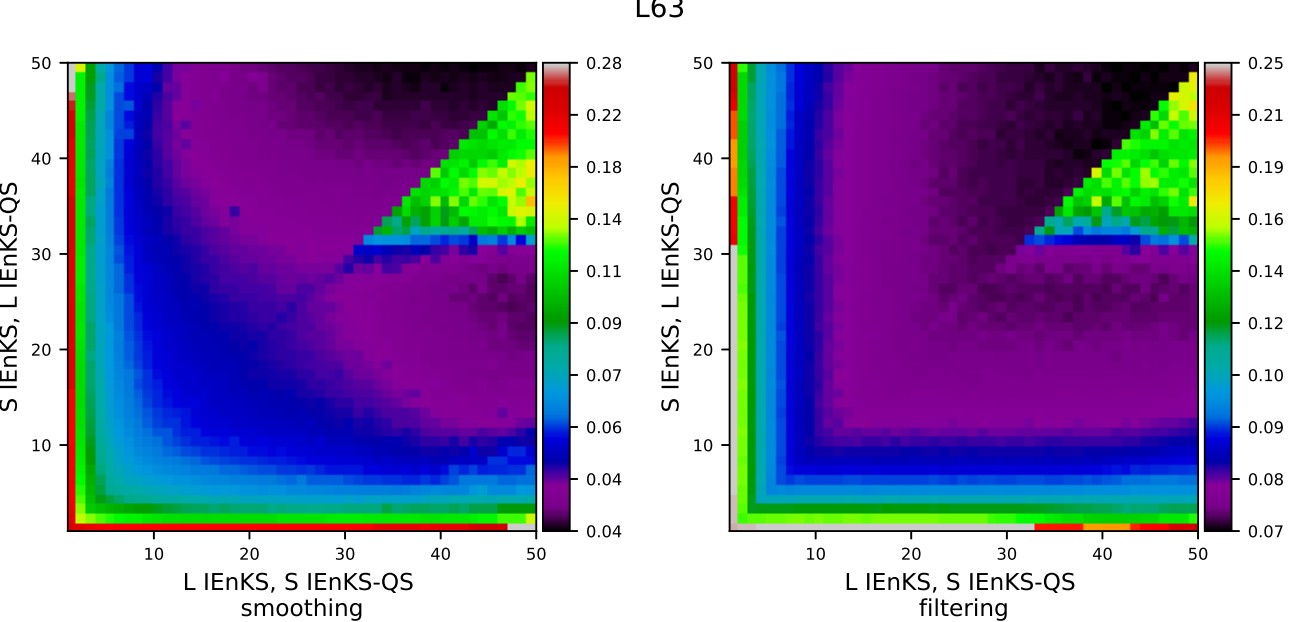

**Figure 9.** IEnKS (lower triangles) and IEnKS-QS (upper triangles, $N_Q = S$) smoothing and filtering aRMSEs as a function of $L$ and $S$ with the L63 model. The $L$ parameter is on the abscissa axis for the IEnKS and on the ordinate axis for the IEnKS-QS. The $S$ parameter is on the abscissa axis for the IEnKS-QS and on the ordinate axis for the IEnKS.

makes up for the difference between 4D-Var and the IEnKS. Both reveal performance improvements with the DAW parameter $S$, which counts the number of observation batches within the DAW. This is a consequence of the Gaussian background approximation: the larger $S$ is, the less the assimilation relies on it.

However, it is observed that this improvement has a limit in the chaotic, perfect model case. The cost function global minimum basin of attraction appears to shrinks with $L$. This causes the Gauss-Newton procedure to miss the cost function global minimum, which deteriorates the assimilation performance.

Quasi-static minimizations lead slowly but surely to the global minimum by repeated cost function minimizations. As the DAW length $L$ is gradually increased, the starting point of the minimization remains in the global minimum basin of attraction. For most $S, L$ couples, the quasi-static IEnKS turns out as a more accurate substitute for the multiple data assimilation IEnKS (IEnKS-MDA).

Unfortunately, this method adds an outer loop which could significantly increase the numerical cost. Precision on intermediate minima being superfluous, one can limit the intermediate Gauss-Newton number of loops. The unavoidable space increments required to minimize the non-quadratic cost function are thus reported in time in the *quasi-convergent* IEnKS.

We did not focus on the applicability of the methods to high-dimensional imperfect models. In particular, we consider very long DAWs, which, even if of high mathematical interest or for low-order reliable models, is less relevant for significantly

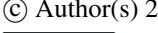



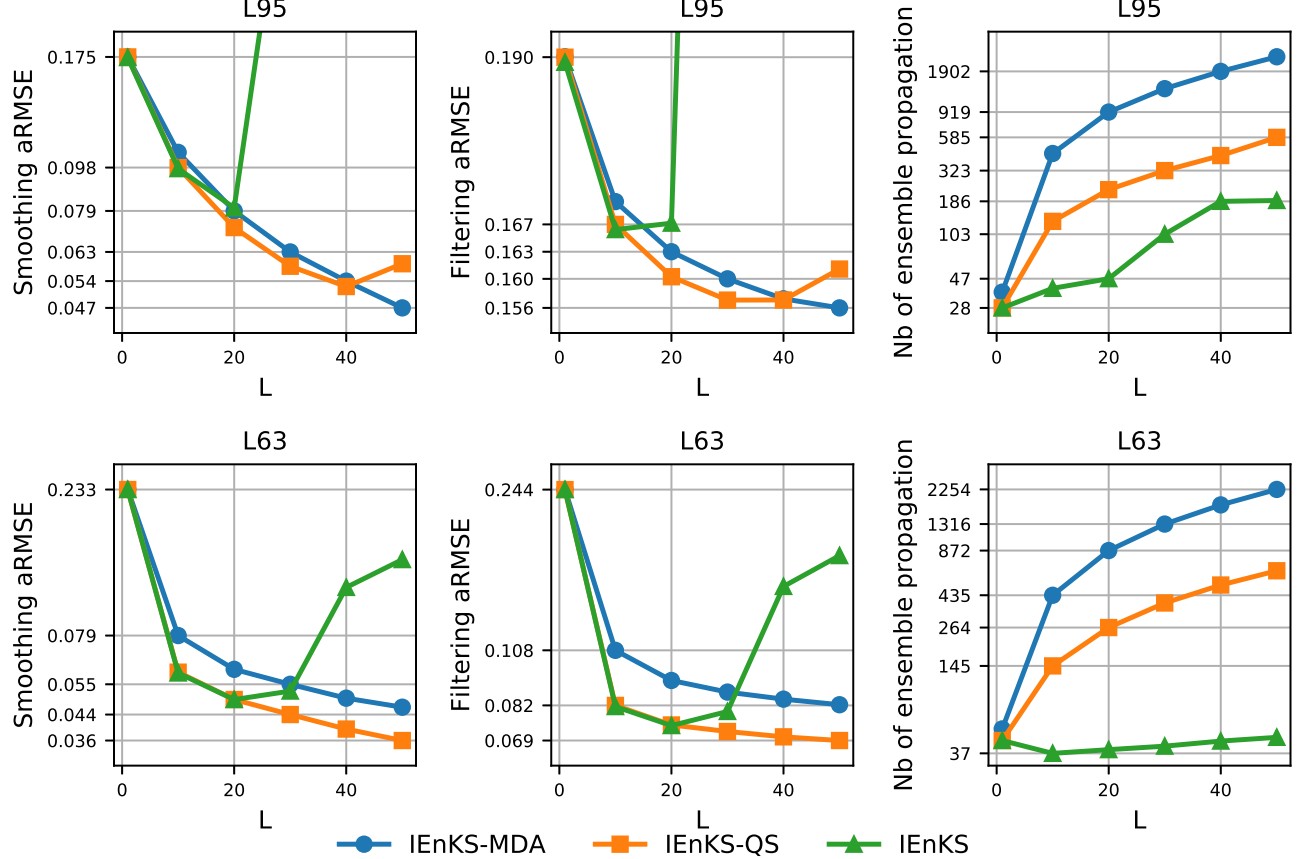

**Figure 10.** IEnKS-MDA ($S = 1$), IEnKS-QS and IEnKS ($S = L$, $N_Q = S$) filtering and smoothing aRMSEs and number of ensemble propagations as a function of $L$ for the L95 and L63 models (logarithmic scale). The ensemble propagations are in units of $\Delta t$. For instance, with $L = 50$, a single ensemble propagation through the DAW counts for 50.

noisy models. An extension to this work would therefore consists in investigating the same ideas but using weak constraint 4D-Var and IEnKS (Trémolet, 2006; Sakov et al., 2017).

## Appendix A: Performance of 4D-Var in the linear, univariate case

The objective of this appendix is to establish a recurrence relation between the 4D-Var eMSE of each cycle. From this relation
5    we will get an expression for the 4D-Var asymptotic eMSE.



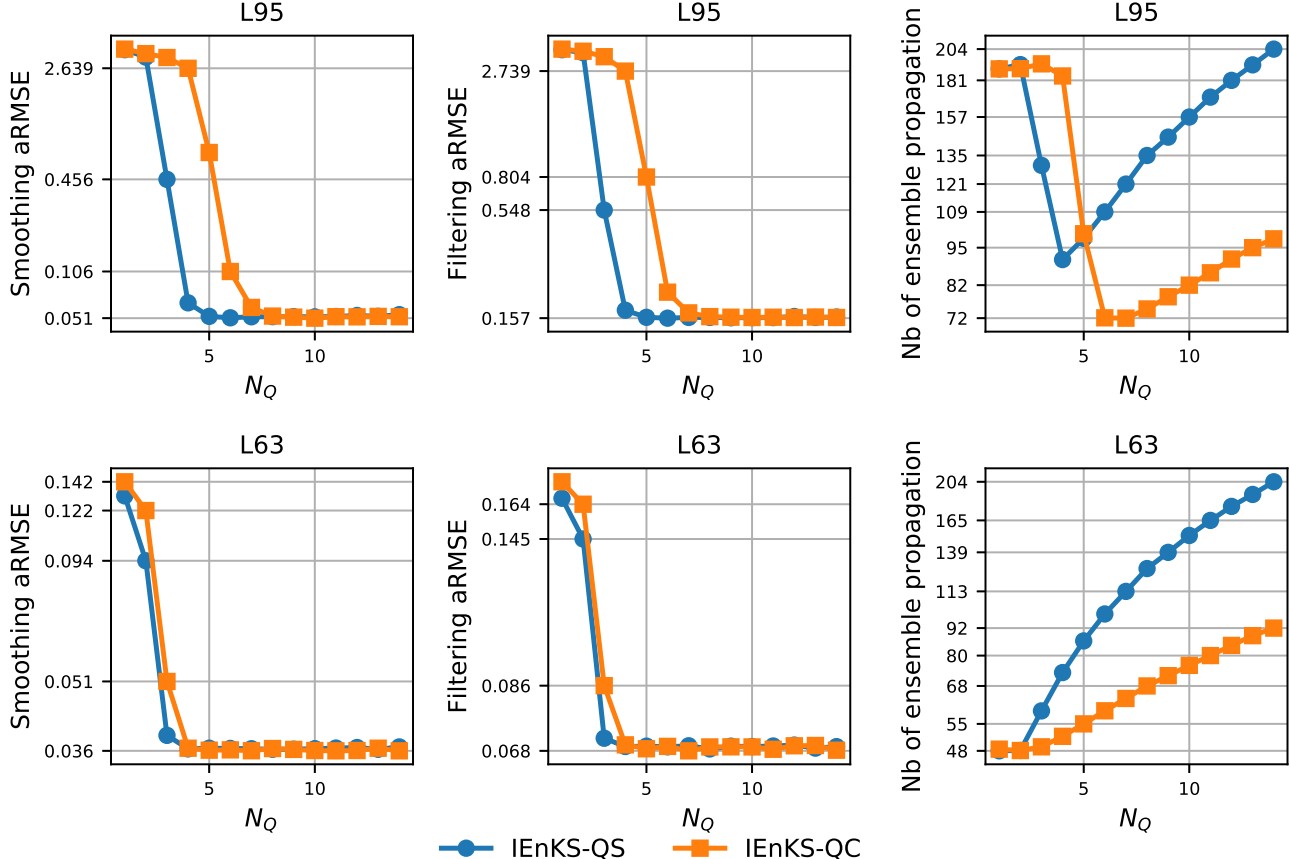

**Figure 11.** IEnKS-QS and IEnKS-QC ($S = L = 50$) filtering and smoothing aRMSEs and number of ensemble propagations as a function of $L$ for the L95 and L63 models (logarithmic scale). The ensemble propagations are in units of $\Delta t$.

We assume $m = 1$ and $\mathcal{M}^l(x) = \alpha^l x$. At the $k$-th cycle, the gradient and Hessian of the 4D-Var cost function Eq. (7) are:

$$\nabla J\left(x_{kS}; y_:, x_{kS}^{\mathrm{b}}\right) = -\frac{1}{b}\left(x_{kS}^{\mathrm{b}} - x_{kS}\right)$$

$$-\frac{h}{r}\sum_{l=K}^{L}\alpha^l\left(y_{kS+l} - h\alpha^l x_{kS}\right), \tag{A1}$$

$$\nabla^2 J\left(x_{kS}; y_:, x_{kS}^{\mathrm{b}}\right) = \frac{1}{b} + \frac{h^2}{r}\sum_{l=K}^{L}\alpha^{2l}, \tag{A2}$$

5  where $y_{kS+L:kS+K}$ is temporarily denoted $y_:$. Because of the operators' linearity, $J$ is quadratic with respect to $x_{kS}$ and convex. Hence, its minimizer $x_{kS}^{\mathrm{a}}$ exists and is characterized by the null gradient equation:

$$\nabla J\left(x_{kS}^{\mathrm{a}}; y_:, x_{kS}^{\mathrm{b}}\right) = 0. \tag{A3}$$





With an exact Taylor expansion around the state $x_{kS}$ we obtain:

$$0 = \nabla J\left(x_{kS}; y_:, x_{kS}^{\mathrm{b}}\right)$$
$$+ \nabla^2 J\left(x_{kS}; y_:, x_{kS}^{\mathrm{b}}\right) \times \left(x_{kS}^{\mathrm{a}} - x_{kS}\right). \tag{A4}$$

Note that $\nabla^2 J\left(x_{kS}; y_:, x_{kS}^{\mathrm{b}}\right)$, given by Eq. (A2), is not random and does not depend on $x_{kS}, y_:, x_{kS}^{\mathrm{b}}$. That is why it is simply

noted $\nabla^2 J$. Using Eqs (1,A4), we have:

$$\nabla^2 J \cdot \left(x_{kS}^{\mathrm{a}} - x_{kS}\right) =$$
$$\frac{1}{b}\left(x_{kS}^{\mathrm{b}} - x_{kS}\right) + \frac{h}{r}\sum_{l=K}^{L}\alpha^l \varepsilon_{kS+l}, . \tag{A5}$$

The random variable $x_{kS}^{\mathrm{b}} - x_{kS} = \alpha^S\left(x_{(k-1)S}^{\mathrm{a}} - x_{(k-1)S}\right)$ is independent from the errors $\varepsilon_{kS+K}, \ldots, \varepsilon_{kS+L}$. Thus, taking the expectation of the square of Eq. (A5) gives the following expression for the eMSE $P_{kS}^{\text{4D-Var}}$ of 4D-Var at time $t_{kS}$:

$$P_{kS}^{\text{4D-Var}} = \mathbb{E}\left[\left(x_{kS} - x_{kS}^{\mathrm{a}}\right)^2\right],$$
$$= \left(\nabla^2 J\right)^{-2}\left(\frac{\alpha^{2S}}{b^2}P_{(k-1)S}^{\text{4D-Var}} + \frac{h^2}{r}\sum_{l=K}^{L}\alpha^{2l}\right), \tag{A6a}$$

$$P_{-S}^{\text{4D-Var}} = b. \tag{A6b}$$

Introducing the notation:

$$\Sigma_K^L = \frac{h^2}{r}\sum_{l=K}^{L}\alpha^{2l} = \frac{h^2}{r}\alpha^{2K}\frac{\alpha^{2S}-1}{\alpha^2-1}, \tag{A7}$$

$$\Delta = \frac{\alpha^{2S}}{\left(1+b\Sigma_K^L\right)^2}, \tag{A8}$$

we obtain

$$P_{kS}^{\text{4D-Var}} = \Delta P_{(k-1)S}^{\text{4D-Var}} + \frac{\Sigma_K^L}{\left(\frac{1}{b}+\Sigma_K^L\right)^2}. \tag{A9}$$

Thus, $\left(P_{kS}^{\text{4D-Var}}\right)_k$ is an arithmetico-geometric sequence. Its limit $P_{\infty S}^{\text{4D-Var}}$ depends on the value of $\Delta$ in the following manner:

$$P_{\infty S}^{\text{4D-Var}} = \begin{cases} \infty & \text{if } \Delta \geq 1, \\ \frac{b^2\Sigma_K^L}{\alpha^{2S}}\frac{\Delta}{1-\Delta} & \text{otherwise.} \end{cases} \tag{A10}$$

The generalization to any asymptotic eMSE with lag $L-l$ is straightforward:

$$P_{\infty S+l}^{\text{4D-Var}} = \begin{cases} \infty & \text{if } \Delta \geq 1, \\ \frac{b^2\Sigma_K^L}{\alpha^{2(S-l)}}\frac{\Delta}{1-\Delta} & \text{otherwise.} \end{cases} \tag{A11}$$

In the multivariate, diagonal case the algebra can be conducted on each direction independently. The eMSE in this case is the sum of the univariate eMSEs of each direction.



## Appendix B: Performance of the IEnKS in the linear, univariate case

The objective of this appendix is to establish a recurrence relation between the IEnKS eMSE of each cycle. From this relation we will get an expression for the IEnKS asymptotic eMSE.

First, it is proved by recurrence that for all $k \geq 0$, $G\left(x_{kS}|y_{kS+L:K}\right)$ is Gaussian with moments:

$$\mathbb{E}\left[x_{kS}|y_{kS+L:K}\right] = \bar{x}_{kS}^{\mathrm{a}}, \tag{B1}$$

$$\mathbb{V}\left[x_{kS}|y_{kS+L:K}\right] = \left(X_{kS}^{\mathrm{a}}\right)^2, \tag{B2}$$

where $\mathbb{V}$ is the variance of a random variable and $\bar{x}_{kS}^{\mathrm{a}}, X_{kS}^{\mathrm{a}}$ are defined by Eqs (15,16). Because of the assumptions Eqs (24c,24d) with Eq. (13) one gets:

$$G\left(x_0|y_{L:K}\right) = J\left(w_0; y_{L:K}, \mathbf{E}_0^{\mathrm{b}}\right) + c_0, \tag{B3}$$

where $c_0$ is a constant independent from $x_0$ and $w_0$. Hence, $G\left(x_0|y_{L:K}\right)$ is Gaussian and its moments are given by Eq. (B1,B2) with $k = 0$. Now, assume $G\left(x_{kS}|y_{kS+L:K}\right)$ is Gaussian with moments given by Eqs (B1,B2) for a $k \geq 0$. Because $x_{(k+1)S} = \mathcal{M}^S\left(x_{kS}\right)$ with $\mathcal{M}$ affine, one gets

$$\mathbb{E}\left[x_{(k+1)S}|y_{kS+L:K}\right] = \mathcal{M}^S\left(\bar{x}_{kS}^{\mathrm{a}}\right) = \bar{x}_{(k+1)S}^{\mathrm{b}}, \tag{B4}$$

$$\mathbb{V}\left[x_{(k+1)S}|y_{kS+L:K}\right] = \alpha^{2S}\left(X_{kS}^{\mathrm{a}}\right)^2 = X_{(k+1)S}^{\mathrm{b}}, \tag{B5}$$

using the conditional expectation properties and Eqs (24a,24b). This result together with Eq. (6) and assumption Eq. (13) yields:

$$G\left(x_{(k+1)S}|y_{(k+1)S+L:K}\right) = \\ J\left(w_{(k+1)S}; y_{(k+1)S+L:(k+1)S+K}, \mathbf{E}_{(k+1)S}^{\mathrm{b}}\right) + c_{k+1}, \tag{B6}$$

where $c_{k+1}$ is a constant independent from $x_{(k+1)S}$ and $w_{(k+1)S}$. Hence $G\left(x_{(k+1)S}|y_{(k+1)S+L:K}\right)$ is Gaussian with moments given by Eqs (B1,B2).

The conditional variance Eq. (B2) is therefore related to the IEnKS performance by the total law of expectation:

$$\begin{aligned}
P_{kS}^{\mathrm{IEnKS}} &= \mathbb{E}\left[\left(\bar{x}_{kS}^{\mathrm{a}} - x_{kS}\right)^2\right], \\
&= \mathbb{E}\left[\mathbb{E}\left[\left(\bar{x}_{kS}^{\mathrm{a}} - x_{kS}\right)^2 |y_{kS+L:K}\right]\right], \\
&= \mathbb{E}\left[\mathbb{V}\left[x_{kS}|y_{kS+L:K}\right]\right], \\
&= \mathbb{E}\left[\left(X_{kS}^{\mathrm{a}}\right)^2\right].
\end{aligned} \tag{B7}$$



Then, from Eq. (16) we get the recurrence relation:

$$\left(X^{\mathrm{a}}_{(k+1)S}\right)^{-2} = \alpha^{-2S}\left(X^{\mathrm{a}}_{kS}\right)^{-2} + \frac{h^2}{r}\sum_{l=K}^{L}\alpha^{2l}, \tag{B8a}$$

$$\left(X^{\mathrm{a}}_{0}\right)^{-2} = \frac{1}{b} + \frac{h^2}{r}\sum_{l=K}^{L}\alpha^{2l}. \tag{B8b}$$

Thus, $X^{\mathrm{a}}_{kS}$ is not random and $P^{\mathrm{IEnKS}}_{kS} = \left(X^{\mathrm{a}}_{kS}\right)^2$. This last equation with Eq. (B8a) tell that the sequence of inverse IEnKS
eMSEs is arithmetico-geometric:

$$\left(P^{\mathrm{IEnKS}}_{kS}\right)^{-1} = \alpha^{-2S}\left(P^{\mathrm{IEnKS}}_{(k-1)S}\right)^{-1} + \Sigma^L_K, \tag{B9a}$$

$$\left(P^{\mathrm{IEnKS}}_{-S}\right)^{-1} = b^{-1}, \tag{B9b}$$

where the notation Eq. (A7) has been used. Properties of arithmetico-geometric sequences allow to obtain the IEnKS asymptotic eMSE:

$$P^{\mathrm{IEnKS}}_{\infty S} = \begin{cases} 0 & \text{if } |\alpha| \leq 1, \\ \frac{r}{h^2\alpha^{2L}}\frac{\alpha^2-1}{\alpha^2} & \text{otherwise,} \end{cases} \tag{B10}$$

and the generalization to asymptotic eMSEs with lag $L-l$ is straightforward:

$$P^{\mathrm{IEnKS}}_{\infty S+l} = \begin{cases} 0 & \text{if } |\alpha| \leq 1, \\ \frac{r}{h^2\alpha^{2(L-l)}}\frac{\alpha^2-1}{\alpha^2} & \text{otherwise.} \end{cases} \tag{B11}$$

Let us now show that the IEnKS eMSE is optimal. Let $x^{\mathrm{a}}_{kS}(y_{kS+L:K})$ be the 4D-Var analysis or any other function of $y_{kS+L:K}$. A bias-variance decomposition (e.g., Bishop, 2006) of this estimator yields:

$$\mathbb{E}_{x_{kS}, y_{kS+L:K}}\left[\left(x_{kS} - x^{\mathrm{a}}_{kS}\right)^2\right]$$

$$= \mathbb{E}_{y_{kS+L:K}}\left[\mathbb{V}_{x_{kS}}\left[x_{kS}|y_{kS+L:K}\right]\right]$$

$$\quad + \mathbb{E}_{y_{kS+L:K}}\left[\left(\mathbb{E}_{x_{kS}}\left[x_{kS}|y_{kS+L:K}\right] - x^{\mathrm{a}}_{kS}\right)^2\right]. \tag{B12}$$

Replacing the moments with Eqs (B1,B2) yields:

$$P^{\mathrm{4D\text{-}Var}}_{kS} = P^{\mathrm{IEnKS}}_{kS} + \mathbb{E}\left[\left(\bar{x}^{\mathrm{a}}_{kS} - x^{\mathrm{a}}_{kS}\right)^2\right].$$

$$\geq P^{\mathrm{IEnKS}}_{kS}. \tag{B13}$$

In the multivariate, diagonal case the algebra can be conducted on each direction independently. Thus, the eMSE in this case is the sum of the univariate eMSEs of each direction.



## Appendix C: Expression of the averaged cost function

The IEnKS averaged cost function $J_{\infty S}$ is the $N$ goes to $\infty$ limit of:

$$\frac{1}{N} \sum_{k=0}^{N-1} J\left(\mathbf{w}; \mathbf{y}_{kS+L:kS+K}, \mathbf{E}_{kS}^{\mathrm{b}}\right) = \frac{1}{2}\|\mathbf{w}\|^2 +$$

$$\frac{1}{2N} \sum_{l=K}^{L} \sum_{k=0}^{N-1} \left\|\mathbf{y}_{kS+l} - \mathcal{H} \circ \mathcal{M}^l\left(\bar{\mathbf{x}}_{kS}^{\mathrm{b}} + \mathbf{X}_{kS}^{\mathrm{b}}\mathbf{w}\right)\right\|_{\mathbf{R}^{-1}}^2. \tag{C1}$$

Expanding the squared norm around $\mathcal{H} \circ \mathcal{M}^l(\mathbf{x}_{kS})$ using Eq. (1) gives:

$$\frac{1}{2} \left\|\mathbf{y}_{kS+l} - \mathcal{H} \circ \mathcal{M}^l\left(\bar{\mathbf{x}}_{kS}^{\mathrm{b}} + \mathbf{X}_{kS}^{\mathrm{b}}\mathbf{w}\right)\right\|_{\mathbf{R}^{-1}}^2 =$$

$$\frac{1}{2}\|\varepsilon_{kS+l}\|_{\mathbf{R}^{-1}}^2 + \frac{1}{2}\|\delta\mathbf{y}_{kS+l}\|_{\mathbf{R}^{-1}}^2 + \varepsilon_{kS+l}^{\mathrm{T}}\mathbf{R}^{-1}\delta\mathbf{y}_{kS+l}, \tag{C2}$$

where $\delta\mathbf{y}_{kS+l} = \mathcal{H} \circ \mathcal{M}^l(\mathbf{x}_{kS}) - \mathcal{H} \circ \mathcal{M}^l\left(\bar{\mathbf{x}}_{kS}^{\mathrm{b}} + \mathbf{X}_{kS}^{\mathrm{b}}\mathbf{w}\right)$. We assume there exists random variables $(\varepsilon_{\infty S}, \mathbf{x}_{\infty S}, \mathbf{E}_{\infty S}^{\mathrm{a}})$ whose distribution is invariant and ergodic with respect to the shift transformation:

$$T: \left(\varepsilon_{kS}, \mathbf{x}_{kS}, \mathbf{E}_{kS}^{\mathrm{a}}\right) \mapsto$$

$$\left(\varepsilon_{(k+1)S}, \mathbf{x}_{(k+1)S}, \mathbf{E}_{(k+1)S}^{\mathrm{a}}\right). \tag{C3}$$

Then because the $(\varepsilon_{kS})_k$ are mutually independent, independent from the $(\mathbf{x}_{kS}, \mathbf{E}_{kS}^{\mathrm{a}})_k$ and identically distributed, $p\left(\varepsilon_{\infty S}, \mathbf{x}_{\infty S}, \mathbf{E}_{\infty S}^{\mathrm{a}}\right) = p(\varepsilon_0) p(\mathbf{x}_{\infty S}, \mathbf{E}_{\infty S}^{\mathrm{a}})$. By Birkhoff's ergodic theorem (see Walters, 1982) we get:

$$\frac{1}{N} \sum_{k=0}^{N-1} \|\varepsilon_{kS+l}\|_{\mathbf{R}^{-1}}^2 \underset{N\to\infty}{\longrightarrow} \mathbb{E}\left[\|\varepsilon_0\|_{\mathbf{R}^{-1}}^2\right] = d, \tag{C4}$$

$$\frac{1}{N} \sum_{k=0}^{N-1} \varepsilon_{kS+l}^{\mathrm{T}}\delta\mathbf{y}_{kS+l} \underset{N\to\infty}{\longrightarrow} \mathbb{E}\left[\varepsilon_0^{\mathrm{T}}\right]^{\mathrm{T}} \mathbb{E}[\delta\mathbf{y}_{\infty S+l}] = 0, \tag{C5}$$

$$\frac{1}{N} \sum_{k=0}^{N-1} \|\delta\mathbf{y}_{kS+l}\|_{\mathbf{R}^{-1}}^2 \underset{N\to\infty}{\longrightarrow} \mathbb{E}\left[\|\delta\mathbf{y}_{\infty S+l}\|_{\mathbf{R}^{-1}}^2\right], \tag{C6}$$

where $\delta\mathbf{y}_{\infty S+l} = \mathcal{H} \circ \mathcal{M}^l(\mathbf{x}_{\infty S}) - \mathcal{H} \circ \mathcal{M}^l\left(\bar{\mathbf{x}}_{\infty S}^{\mathrm{b}} + \mathbf{X}_{\infty S}^{\mathrm{b}}\mathbf{w}\right)$ and $\bar{\mathbf{x}}_{\infty S}^{\mathrm{b}}, \mathbf{X}_{\infty S}^{\mathrm{b}}$ are respectively the mean and normalized anomaly of $\mathcal{M}^S(\mathbf{E}_{\infty S}^{\mathrm{a}})$. Finally,

$$J_{\infty S}(\mathbf{w}) = \frac{1}{2}\|\mathbf{w}\|^2 + \frac{dS}{2} + \frac{1}{2} \sum_{l=K}^{L} \mathbb{E}\left[\|\delta\mathbf{y}_{\infty S+l}\|_{\mathbf{R}^{-1}}^2\right]. \tag{C7}$$

*Acknowledgements.* The authors are grateful to S. Gürol for her comments and suggestions on the manuscript. CEREA is a member of Institut Pierre-Simon Laplace (IPSL).



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
