# Peer review of "Quasi static ensemble variational data assimilation: a theoretical and numerical study with the iterative ensemble Kalman smoother"

_Nonlinear Processes in Geophysics, 2017_

## Referee Comment (RC1) · C. Pires (Referee) · 15 Dec 2017

The authors have performed a thorough study of the applicability of Quasi-Static (QS) variational data assimilation schemes on an ensemble Kalman smoother (EnKS), leading to the iterative ensemble Kalman smoother (IEnKS). The QS algorithm has been shown to be crucial by Pires et al. (1996) in 4DVar DA on nonlinear chaotic models. That procedure keeps the analysis error within the region of validity of the linear tangent approach, which nearly corresponds to the cost-function attraction basin including the true model state.

The manuscript is very well written, structured, rigorous, and presents some novel results, being thus well suited for publication in NPG. There are only a few aspects

which would be worth to mention or discuss which could still improve the manuscript.

1 In section 3 (Quasi Static algorithms) it is worth to mention and to put into context the 'Sequential Quasi Static Variational Assimilation' (section 4.2 of (Pires et al. 1996)) as a variation of the QS scheme. There, the QS scheme is only applied to the very beginning DAW. Then, as time progress, the long DAWs move forward by small steps but at the expense of overlapping with previous DAWs. In the subsequent DAWs, a single 4DVAR-DA is performed using the first guess provided by the DA issued from the previous DAW. Therefore, in a sequential DA scheme the QS scheme is not necessary if a substantial overlapping of the DAWs holds (observations assimilated multiple times). However, the cost may be larger or comparable to QS with large jumps of the DAW.

2 In the discussion of upper triangles of Figs. 8 (L95) and 9 (L63), showing the average smoothing and filtering errors, the authors should discuss how far it is useful to increase the DAW length. In Pires et al (1996), it is presented the concept of useful assimilation window $\sim$ -ln(0.01/(2 Lambda-max), beyond which the DA is not useful anymore where Lambda-max is the Largest Lyapunov value. Giving the steps delta-t and lambda-max, the authors may provide the largest useful DAW length Lmax.

3 In the discussion and conclusions, the authors should add a small paragraph on the limitations of extending the DAW length in DA with nonperfect models (refer to Swanson et al 1998).

Carlos Pires

---

## Referee Comment (RC2) · Anonymous Referee #2 · 19 Jan 2018

——————————————- General Comments ——————————————-

The aims of this paper is to look at the application of the quasi-static scheme developed by Pires et al (1994) to ensemble variational assimilation algorithms. This extension is useful for large data assimilation windows with chaotic non-linear models, as standard approaches fail to find the global minimum of the cost function.

The paper is restricted to the case of low order models with perfect model assumption. This limitation seems restrictive and this clearly diminishes the scope of the results. Indeed, the original papers on quasi-static variational assimilation (QSVA) were at least partly addressing the case of higher dimensions and model error (e.g., Swanson et al. 1998).

[Figure]

The application of QSVA together with ensemble formulation has received some interest in the community (e.g., Goodlif et al. 2015) but the paper is in my opinion missing a discussion on how the results compare with the one of Goodlif et al. 2015.

The paper is sometimes hard to follow : in a first section, theoretical developments are used to compare the performance of 4D-Var and of the IenKS in a linear and highly simplified context. This is interesting but then 4D-Var is dropped out of the DA schemes that are considered and it is not obvious why. The limitations of the standard IEnKS with increasing DA windows are well illustrated and lead to section 3 with quasi-static versions of the IEnKS compared to standard ones. Here, a novel algorithm is discussed, the MDA. I would recommend to focus on IEnKS only; dropping the 4D-Var and the MDA versions to make the paper more focussed.

Figures are generally clear, with the exception of Fig. 10 and 11 where the third panel (about the number of ensemble propagations) is put on the same "level" as he two other ones (RMSE) which is confusing at first glance. There is yet a general problem with the colours that do not render well in gray scale and thus likely confusing for colour blinded people : the authors may consider using better colour maps for this purpose.

Overall, the paper may be suitable to publication only if those concerns are properly dealt with, which is why I would recommend a major revision.

———————————————- Specific Comments ———————————————-

[1] I would recommend that the paper is more clear about the limitations of the study. It is definitively in the text but not in the title and in the abstract. I would mention the perfect model assumption in the abstract. Also, the title is too general. I would make it more specific, for instance "Performances of the quasi-static formulation of the iterative Kalman Smoother on low-order models", or "A quasi-static version of the strong constraint iterative Kalman Smoother" for instance. [2] Page 2, line 18. Is it a known fact that the number of local minima increases exponentially with the data assimilation window ? If yes, please provide a proof or quote, if not please be more

vague.

[3] Page 2, lines 20-25. There are other methods that address the convergence of minimization despite the non-linearity of the operators by using globalization methods, even published by the authors (e.g. Preconditioning and globalizing conjugate gradients in dual space for quadratically penalized nonlinear-least squares problems by Gratton et. al.). Please add and comment references with alternative minimization algorithms to address non-linearity.

[3] Page 2, last paragraph. You mention that your paper is designed to be a "more complete analytical and numerical investigation", but you do not comment on the main results of the paper you are citing. Please provide a better discussion of your paper with the existing literature.

[4] Page 3, line 19. Your paper is about low dimensional and perfect model, such that I would change the sentence to "not meant to improve high-dimensional nor imperfect models".

[5] Page 6, line 16 : please detail in which sense the inverse square root of the matrix is taken , as it is ambiguous.

[6] Page 11, line 5 : I do not understand the qualitative explanation that is given, please reformulate. , [7] Page 20 : the description of Figs. 8, 9 and 10 is very short, with only a few lines to comment 10 panels. Please consider discussing more the resultts or simplifying the figures by showing only what you tell.

[8] Page 23 and 24 : I do not understand what the number of ensemble propagation is, and the paper is missing an explanation of why in Fig 11 we observe different behaviours between L63 and L95, and also why it has non-monotonic evolution with parameter NQ.

---

## Author Comment (AC1) · 20 Feb 2018

We are grateful to the Reviewer for his suggestions.

1. **In section 3 (Quasi Static algorithms) it is worth to mention and to put into context the 'Sequential Quasi Static Variational Assimilation' (section 4.2 of (Pires et al. 1996)) as a variation of the QS scheme**

   Indeed. Thank you for pointing this out. We now refer to this *sequential* QSVA scheme in the revised manuscript. Note that this can be seen as an ancestor of the MDA IEnKS. However, as shown in Bocquet and Sakov (2014), a sequential QSVA cannot be transposed directly into an EnVar scheme without further modification because of the multiple assimilation of the same observations, hence the

MDA IEnKS.

2. **In the discussion of upper triangles of Figs. 8 (L95) and 9 (L63), show-
ing the average smoothing and filtering errors, the authors should discuss
how far it is useful to increase the DAW length. In Pires et al (1996), it is
presented the concept of useful assimilation window -ln(0.01/(2 Lambda-
max), beyond which the DA is not useful anymore where Lambda-max is
the Largest Lyapunov value. Giving the steps delta-t and lambda-max, the
authors may provide the largest useful DAW length Lmax.**

The idea of useful data assimilation length is a very nice concept introduced by
Pires et al. (1996). For both low-order models, one obtains $L_{max}^{L95} = \frac{-\ln(0.01)}{2\lambda^{L95}\Delta t^{L95}} = \frac{-\ln(0.01)}{2\times1.7\times0.05} \simeq 27$ and $L_{max}^{L63} = \frac{-\ln(0.01)}{2\lambda^{L63}\Delta t^{L63}} = \frac{-\ln(0.01)}{2\times0.91\times0.02} \simeq 127$. This concept is now
recalled (twice) in the revised manuscript with a reference to Pires et al. (1996).

Note, however, that it has some limitations. First, it applies to the filtering error (at
present time), not to the smoothing error – at least not directly. Second, this result
does not easily translates to an advanced cycled scheme such as the IEnKS$_{QS}$,
where a lot of observations have already been assimilated and their information
condensed in the background. Thus, the performance gain with the DAW length
comes from the precision of this Gaussian background approximation – a preci-
sion that the linearized theory is not able to evaluate. We have shown in Bocquet
and Sakov (2013, 2014) and in the present manuscript, that one can go very far
in the past – well beyond $27\Delta t$ in the L95 case – and still improve the smoothing
RMSE. Third, this useful length does not account for nonlinearities, the appear-
ance of local minima, and correlatively potential saturation. As a result there is a
somehow arbitrary constant in its definition ($0.01$ here). In Pires et al. (1996), it is
related to the targeted error.

The length that we estimate in Sec. 2.4 can be seen to some degree as an
improvement on Pires et al. (1996)' endeavor of a useful length by estimating

the constant resorting to saturation and the occurence of local minima in the cost function.

3. **In the discussion and conclusions, the authors should add a small paragraph on the limitations of extending the DAW length in DA with nonperfect models (refer to Swanson et al 1998).**

Thank you for the suggestion. Extending the DAW length is less relevant for significantly noisy models. Swanson et al. (1998) showed that the perfect model results are expected to extend to the imperfect model case provided that the growth rate of the model error is similar to that of the leading Lyapunov vectors of the system. This is discussed in the revised manuscript at the end of the conclusion and a reference to Swanson et al. (1998) has been added.

**References**

Bocquet, M., Sakov, P., 2013. Joint state and parameter estimation with an iterative ensemble Kalman smoother. Nonlin. Processes Geophys. 20, 803–818. doi:10.5194/npg-20-803-2013.

Bocquet, M., Sakov, P., 2014. An iterative ensemble kalman smoother. Q. J. R. Meteorol. Soc. 140, 1521–1535. doi:10.1002/qj.2236.

Pires, C., Vautard, R., Talagrand, O., 1996. On extending the limits of variational assimilation in nonlinear chaotic systems. Tellus A 48, 96–121. doi:10.3402/tellusa.v48i1.11634.

Swanson, K., Vautard, R., Pires, C., 1998. Four-dimensional variational assimilation and predictability in a quasi-geostrophic model. Tellus A 50, 369–390. doi:10.1034/j.1600-0870.1998.t01-4-00001.x.

---

## Author Comment (AC2) · 20 Feb 2018

We thank the Reviewer for the questions and comments. There are a few points on which we partly or totally disagree and we justify why.

**General comments**

1. **The paper is restricted to the case of low order models with perfect model assumption. This limitation seems restrictive and this clearly diminishes the scope of the results. Indeed, the original papers on quasi-static varia-tional assimilation (QSVA) were at least partly addressing the case of higher**

[Figure]

**dimensions and model error (e.g., Swanson et al. 1998).**

We disagree in two ways:

(a) First, two thirds of the paper are on the theory of QSVA in an EnVar context, whose scope is broad, and significantly larger than numerically testing QSVA with EnVar/hybrid methods as in Bocquet and Sakov (2013, 2014); Goodliff et al. (2015). This is independent from the dimension of the problem, though it depends on the perfect-model assumption.

(b) Secondly, the reviewer seems to assume that data assimilation methods based on perfect model assumptions cannot be applied to imperfect models. This is clearly not the case since strong-constraint 4D-Var has been applied in operational meteorological forecast for 20 years to imperfect models. Hence, of course, the algorithm proposed in this manuscript can be applied to imperfect models as well, with limitations that have been discussed in Swanson et al. (1998). Although it is important to mention this point, we consider it a rather distinct subject from our endeavor in the theory part of this manuscript.

Note, that a mathematically consistent variant of the IEnKF/IEnKS with additive model error has been recently designed and tested (Sakov and Bocquet, 2018; Sakov et al., 2018), so that we could contemplate in a near future an extension of the present study to an IEnKS where model error is properly accounted for.

2. **The application of QSVA together with ensemble formulation has received some interest in the community (e.g., Goodlif et al. 2015) but the paper is in my opinion missing a discussion on how the results compare with the one of Goodlif et al. 2015.**

   To our knowledge, the first quasi-static algorithm in an EnVar/hybrid context has been proposed and tested in Bocquet and Sakov (2013, 2014), specifically the

MDA IEnKS scheme. Another attempt came from M. Jardak and O. Talagrand at about the same time but reported in conferences, and it was only concerned with 4D-Var as it was applied to a *non-cycled* EDA scheme. The interested reader can have a look at their very recent 2018 submission in Nonlinear Processes in Geophysics.

Goodlif et al. 2015 provides a numerical exploration of the impact of flow dependent background covariances and QS minimizations on the performance of hybrid schemes with the L63 model. QSVA is merely used as a tool following Pires et al. (1996). This impact of QSVA is just established on numerical experiments, which confirm the findings of Pires et al. (1996), or those of Bocquet and Sakov (2013, 2014) with the MDA IEnKS. It does not seem that there is much more to mention, as far as QSVA is concerned.

Here, by contrast, our goal is to justify theoretically and give insights about QSVA in the context of cycled EnVar data assimilation. This is later illustrated by algorithms and numerics.

A more thourough (but not really necessary in our opinion) would be for instance to compare our L63 numerical results with those of Goodliff et al. (2015):

(a) First, the critical and interesting Sec. 3.6 of Goodliff et al. (2015) is not sufficiently documented. For instance, we do not know if the algorithms are cycled or just averages over several instances. The definition of their RMSE Eq. (29) does tell the reader how the average is actually done (something must be missing in the definition), and it mixes filtering and smoothing RMSEs which makes any interpretation more difficult.

(b) Second, Goodliff et al. (2015) showed that the ETKS outperforms all the schemes in their study. Since the IEnKS systematically outperforms the ETKS in all conditions (and in particular L63) as long as the DAW length is not overwhelmingly long (for a chaotic model), then one concludes that our

RMSEs would be systematically equal or smaller that those reported for any hybrid scheme in Goodliff et al. (2015).

We have increased the discussion/comparison on Bocquet and Sakov (2013, 2014); Goodliff et al. (2015), and made a more detailed reference to those in the introduction of the revised manuscript.

3. **The paper is sometimes hard to follow : in a first section, theoretical developments are used to compare the performance of 4D-Var and of the IEnKS in a linear and highly simplified context. This is interesting but then 4D-Var is dropped out of the DA schemes that are considered and it is not obvious why. The limitations of the standard IEnKS with increasing DA windows are well illustrated and lead to section 3 with quasi-static versions of the IEnKS compared to standard ones. Here, a novel algorithm is discussed, the MDA. I would recommend to focus on IEnKS only; dropping the 4D-Var and the MDA versions to make the paper more focused.**

The discussion about 4D-Var is here to illustrate the impact of an improper modeling of the prior pdf in a simplified linear context. The analogy with the improper prior modeling of the IEnKS in a non-linear context becomes then clearer. The 4D-Var is dropped in the numerical experiments because the proof that ensemble variational methods are numerically more efficient than variational methods has already been established (Bocquet and Sakov, 2013).

That is why 4D-Var is replaced by another quasi-static ensemble variational method: the MDA IEnKS. This is not a novel method (Bocquet and Sakov, 2013, 2014), and it is the first documented quasi-static EnVar method (with $S = 1$ at least). Note also that the question of how long the data assimilation should or could be in an EnVar context that we addressed in this paper was first formulated in Bocquet and Sakov (2014) and discussed in their conclusion as an open question.

Hence, we are not convinced that the manuscript would benefit from your present suggestions.

4. **Figures are generally clear, with the exception of Fig. 10 and 11 where the third panel (about the number of ensemble propagations) is put on the same "level" as he two other ones (RMSE) which is confusing at first glance. There is yet a general problem with the colours that do not render well in gray scale and thus likely confusing for colour blinded people : the authors may consider using better colour maps for this purpose.**

   Thank you very much for the suggestion. In the revised manuscript, we choose to use a colormap that renders properly in grayscale. To keep color variability with small RMSEs, values beyond a certain limit have the same color. Also, each axis of Fig. 10, 11 has its own title to avoid confusion.

**Specific comments**

1. **I would recommend that the paper is more clear about the limitations of the study. It is definitely in the text but not in the title and in the abstract. I would mention the perfect model assumption in the abstract. Also, the title is too general. I would make it more specific, for instance "Performances of the quasi-static formulation of the iterative Kalman Smoother on low-order models", or "A quasi-static version of the strong constraint iterative Kalman Smoother" for instance**

   As we explained, we do not believe that the findings of this paper are as limited as you claim they are. That said, we can certainly mention the perfect model assumption in the abstract. We did so in the revised manuscript.

   We believe our title was not too general. But it can surely help the reader to make it more focused. The titles that you propose do not reflect the generality of our

findings. Indeed, the IEnKS is the archetype of a deterministic EnVar method and we use it as such in this manuscript (as derived in Bocquet and Sakov, 2014). We expect any good (or close to optimal) EnVar method to reach the same conclusions.

We believe "Quasi-static ensemble variational data assimilation: a theoretical and numerical study with the iterative ensemble Kalman smoother" now perfectly reflects the content of the manuscript.

2. **Page 2, line 18. Is it a known fact that the number of local minima increases exponentially with the data assimilation window ? If yes, please provide a proof or quote, if not please be more vague**

This statement comes from Swanson et al. (1998) p.377 and is justified by Pires et al. (1996) p.106. A "may" mitigates the statement, since this may have only been proven for emblematic chaotic model (such as the baker map). These references have been added at this point in the revised manuscript. Thank you for this clarification enquiry.

3. **Page 2, lines 20-25. There are other methods that address the convergence of minimization despite the non-linearity of the operators by using globalization methods, even published by the authors (e.g. Preconditioning and globalizing conjugate gradients in dual space for quadratically penalized nonlinear-least squares problems by Gratton et. al.). Please add and comment references with alternative minimization algorithms to address nonlinearity.**

Please check your definition of *globalization methods*. They do not aim at finding the global minimum but are meant to obtain convergence of the iterates for every initial guess. About finding the global minimum, we gave references to Ye et al. (2015); Judd et al. (2004) which are the only one we can think of in the geophysical data assimilation context (besides QSVA).

The reason why methods looking for a global minimum are seldomly used in data assimilation was given p.2 line 12.

4. **Page 2, last paragraph. You mention that your paper is designed to be a "more complete analytical and numerical investigation", but you do not comment on the main results of the paper you are citing. Please provide a better discussion of your paper with the existing literature.**

   The existing literature as far as quasi-static hybrid/EnVar methods are concerned is Bocquet and Sakov (2013, 2014); Goodliff et al. (2015). A discussion is given in our response to question 2 of the general comments, and to some extent included in the revised manuscript.

5. **Page 3, line 19. Your paper is about low dimensional and perfect model, such that I would change the sentence to "not meant to improve high-dimensional nor imperfect models".**

   The sentence has been corrected. Thank you for spotting this mistake.

6. **Page 6, line 16 : please detail in which sense the inverse square root of the matrix is taken , as it is ambiguous.**

   We mention in the revised manuscript that it is the unique symmetric definite positive square root matrix of a symmetric definite positive matrix (which is by far the most common definition).

7. **Page 11, line 5 : I do not understand the qualitative explanation that is given, please reformulate**

   The explanation means that to assimilate the same number of observations, an algorithm using a greater value for the DAW parameter $S$ need less cycles. Because a cost function approximation is made on the background term each cycle, this algorithm relies less often on this approximation making the analysis more

accurate. The sentence has been reformulated in the revised manuscript. Thank you.

8. **Page 20 : the description of Figs. 8, 9 and 10 is very short, with only a few lines to comment 10 panels. Please consider discussing more the results or simplifying the figures by showing only what you tell.**

We fully agree that the discussion was too short. Thank you for pointing out this weakness.

A description of the performance variation with the DAW parameters has been added about Figs. 8, 9. Then the IEnKS$_{QS}$ filtering RMSE invariance with $L$ is discussed and compared to the 4D-Var filtering performance in a linear context. A missing discussion of the performance with L63 has been added as well, about Fig. 10.

9. **Page 23 and 24 : I do not understand what the number of ensemble propagation is, and the paper is missing an explanation of why in Fig 11 we observe different behaviors between L63 and L95, and also why it has non-monotonic evolution with parameter NQ.**

The number of ensemble propagations is the total number of times an ensemble is propagated with a time step of $\Delta t$ in the future, divided by the total number of observations assimilated. The similarity with an Heaviside function comes from the 2 main regimes for the RMSE. When $N_Q$ is too small the methods does not locate the global minimum and the RMSE is close to the climatological variance. When $N_Q$ is sufficiently high, the method locates the global minimum and the RMSE is low. The difference of number of ensemble propagation between the L63 model and the L95 model comes from the minimization. When it misses the global minimum, it does not converge with L95 leading to a large number of iteration and ensemble propagations. It converges with L63 but to a local extremum leading to few iterations and few ensemble propagations.

[Figure]

This discussion has been added to the revised manuscript. Thank you for pointing out to this weakness in the original manuscript.

**References**

Bocquet, M., Sakov, P., 2013. Joint state and parameter estimation with an iterative ensemble Kalman smoother. Nonlin. Processes Geophys. 20, 803–818. doi:`10.5194/npg-20-803-2013`.

Bocquet, M., Sakov, P., 2014. An iterative ensemble kalman smoother. Q. J. R. Meteorol. Soc. 140, 1521–1535. doi:`10.1002/qj.2236`.

Goodliff, M., Amezcua, J., Van Leeuwen, P.J., 2015. Comparing hybrid data assimilation methods on the lorenz 1963 model with increasing non-linearity. Tellus A 67, 26928. doi:`10.3402/tellusa.v67.26928`.

Judd, K., Smith, L., Weisheimer, A., 2004. Gradient free descent: shadowing, and state estimation using limited derivative information. Physica D 190, 153–166. doi:`http://doi.org/10.1016/j.physd.2003.10.011`.

Pires, C., Vautard, R., Talagrand, O., 1996. On extending the limits of variational assimilation in nonlinear chaotic systems. Tellus A 48, 96–121. doi:`10.3402/tellusa.v48i1.11634`.

Sakov, P., Bocquet, M., 2018. Asynchronous data assimilation with the EnKF in presence of additive model error. Tellus A 70, 1414545. doi:`10.1080/16000870.2017.1414545`.

Sakov, P., Haussaire, J.M., Bocquet, M., 2018. An iterative ensemble Kalman filter in presence of additive model error. Q. J. R. Meteorol. Soc. 0, 0–0. doi:`10.1002/qj.3213`. accepted for publication.

Swanson, K., Vautard, R., Pires, C., 1998. Four-dimensional variational assimilation and predictability in a quasi-geostrophic model. Tellus A 50, 369–390. doi:`10.1034/j.1600-0870.1998.t01-4-00001.x`.

Ye, J., Rey, D., Kadakia, N., Eldridge, M., Morone, U., Rozdeba, P., Abarbanel, H., Quinn, J., 2015. Systematic variational method for statistical nonlinear state and parameter estimation. Phys. Rev. E 92, 052901. doi:`10.1103/PhysRevE.92.052901`.

[Figure]

2017-65, 2017.

---

## Author Response (AR2)

**Quasi static ensemble variational data assimilation: a theoretical and numerical study with the iterative ensemble Kalman smoother**

Anthony Fillion[1,2], Marc Bocquet[1], and Serge Gratton[3]

[1]CEREA, Joint Laboratory École des Ponts ParisTech and EDF R&D, Université Paris-Est, Champs-sur-Marne, France
[2]CERFACS, Toulouse, France
[3]INPT-IRIT, Toulouse, France

*Correspondence to:* Anthony Fillion (anthony.fillion@enpc.fr)

Dear Editor,

Please find in this document:

– A point-by-point response to your comments

5    – A manuscript that indicates all the changes and corrections.

Yours sincerely, The authors.

**Response to Editor review of**
**"Quasi static ensemble variational data assimilation: a theoretical and numerical study with the iterative ensemble Kalman smoother" (ID npg-2017-65)**
**by A. Fillion, M. Bocquet and S. Gratton**

A. Fillion, M. Bocquet and S. Gratton

March 14, 2018

**Response to Alberto Carrassi, Editor**

We are grateful to the Reviewer for his suggestions.

**Comments**

1. **Page 2, Line 13: I suggest the authors to add one line to clarify better what they mean by "local" versus "global". While this is pretty known to many, I think a clarification would help readers less familiar with data assimilation techniques. This is also somehow related to the Specific point 3 of Reviewer 2.**

   The terms "local" and "global" are now properly defined, and the paragraph has been improved. Thank you for the suggestion.

2. **Page 2, Line 22: Invert the order of references.**

   The references have been swaped. Thank you for spotting this mistake.

3. **Page 7, Line 17-20: I would suggest the Authors to include the explicit equation for the Hessian (or its unique symmetric definite positive inverse square root) as a function of $\mathbf{w}^{\mathrm{a}}$, $\mathbf{y}$ and $\mathbf{E}^{\mathrm{b}}$. This is also somehow related to the Specific point 6 of Reviewer 2.**

   We did so according to your suggestion. Thank you.

4. **Page 22, Line 13: I think "...L63 and L95..." should be "... L95 and L63 ...".**

   Indeed, you are right. The order has been corrected.

5. **I would further suggest to remove "... merely ..." from Page 22, Line 6, that is not necessary.**

   Thank you; the adverb has been deleted in the revision.

[revised manuscript text omitted]